# *Eurhadina* (*Singhardina*) Mahmood (Hemiptera: Cicadellidae: Typhlocybinae) from China: A Review of the Asian Species with Descriptions of 14 New Species [note 1]

**DOI:** 10.3390/insects13040345

**Published:** 2022-03-31

**Authors:** Juan Zhou, Yalin Zhang, Min Huang

**Affiliations:** Key Laboratory of Plant Protection Resources and Pest Management of the Ministry of Education, Entomological Museum, Northwest A&F University, Xianyang 712100, China; dearzhoujuan@foxmail.com

**Keywords:** Typhlocybini, Typhlocybinae, morphology, distribution, taxonomy

## Abstract

**Simple Summary:**

In this study, 50 species of the leafhopper subgenus *Eurhadina* (*Singhardina*) Mahmood from China are reviewed based on comparative morphological characteristics, including 14 new species, four additional species and two new synonymies. A key to all Chinese *Singhardina* species is also provided.

**Abstract:**

In this study, 50 species of the leafhopper subgenus *Eurhadina* (*Singhardina*) Mahmood from China are reviewed based on comparative morphological characteristics, including 14 new species: *Eurhadina* (*Singhardina*) *amacularis*, *E.* (*S.*) *extensa*, *E.* (*S.*) *flaviscutella*, *E.* (*S.*) *foliiformis*, *E.* (*S.*) *galacta*, *E.* (*S.*) *gracilifurca*, *E.* (*S.*) *lata*, *E.* (*S.*) *parilintanonica*, *E.* (*S.*) *quadrimacularis*, *E.* (*S.*) *recta*, *E.* (*S.*) *scalesa*, *E.* (*S.*) *scamba*, *E.* (*S.*) *scandens* and *E.* (*S.*) *uprotrusa*
**sp. nov.** Four additional species *E.* (*S.*) *fasciata*, *E.* (*S.*) *jarrayi*, *E.* (*S.*) *prima* and *E.* (*S.*) *zadyma* are recorded from China for the first time. Two new synonymies are proposed. *Eurhadina* (*Singhardina*) *flavescens* Huang et Zhang, 1999 **syn. nov.** is synonymized with *Eurhadina wuyiana* Yang et Li, 1991 and *Eurhadina rubromia* Cai et Kuoh, 1993 **syn. nov.** is synonymized with *Eurhadina* (*Singhardina*) *biavis* Yang et Li, 1991. A key to all Chinese *Singhardina* species is also provided.

## 1. Introduction

The leafhopper genus *Eurhadina* Haupt, 1929 belongs to the subfamily Typhlocybinae within the family Cicadellidae (Hemiptera: Cicadomorpha) and is distributed in the Palearctic and Oriental regions. Its type species is *Cicada pulchella* Fallen, 1806 from Sweden [1]. *Eurhadina* includes three subgenera: *Eurhadina* (*Eurhadina*) Haupt 1929, *Eurhadina* (*Singhardina*) Mahmood 1967, and *Eurhadina* (*Zhihadina*) Yang & Li 1991, which includes only one species from China. *Eurhadina* now has 40 species in China with China now having the richest species diversity of this genus.

The subgenus *Singhardina* was proposed by Dworakowska [2]. Subsequently, five species in China, Singapore, India, and Vietnam were transferred from the nominate subgenus to it.

Until now, *Singhardina* comprised 68 described species, which were divided into six groups by Dworakowska according to morphological characteristics [3], in which 34 species from China belong to four species groups: *E.* (*S.*) *punjabensis*, *E.* (*S.*) *vittata*, *E.* (*S.*) *mamata*, and *E.* (*S.*) *robusta*. Recently, based on molecular evidence, Zhang and Huang [4] proposed the new *E.* (*S.*) *rubra* species group, separating it from the *E.* (*S.*) *punjabensis* group and placing the *E.* (*S.*) *vittata* group into the *E.* (*S.*) *punjabensis* group. However, the combination of the *E.* (*S.*) *vittata* group and the *E. (S.) punjabensis* group had low bootstrap values in the MP and ML trees and was in need of further testing. In the present study, we still keep the *E.* (*S.*) *vittata* group as an independent group while treating the *E.* (*S.*) *rubra* group as a separate species group.

In this study, 14 new species and related known species are illustrated and photographed, four additional species are recorded for the first time from China, and two new synonymies are proposed. An identification key to 50 species and a checklist to each species group of the subgenus *Singhardina* from China are provided.

## 2. Materials and Methods

The type specimens of the new species are deposited in the collections of the Entomological Museum, Northwest A&F University, Yangling, China. The entire abdomens of the examined male specimens were dissected and soaked in cold 10% NaOH solution overnight, subsequently rinsed for 30 s with pure water, and then transferred to glycerine for further preservation. Habitus photographs were taken with an automontage QIMAGING Retiga 4000R digital camera (CCD) (QImaging, Surrey, BC, Canada). Line diagrams of the male genitalia were drawn under an OLYMPUS PM-10AD microscope (Olympus, Tokyo, Japan). All figures were edited using Adobe Photoshop CS 6.0 (Adobe Systems, San Jose, CA, USA). The terminology of the adult genitalia generally follows Zhang [5] except for wing venation, which follows Dworakowska [6]. The map was obtained from Google Maps @2020 (http://www.gditu.net/, accessed on 15 March 2022).

## 3. Results and Discussion

### 3.1. Generic Character

*Eurhadina* (*Singhardina*) Mahmood, 1967

*Eurhadina* (*Singhardina*) Mahmood, 1967: 32. Type species: *Singhardina robusta* Mahmood, 1967.

**Description.** (Modified from Dworakowska, 2002 [3]; Mahmood, 1967 [7])

Small species, 2–4 mm in length (including wing). Body depressed and robust with crown to a face narrowly rounded in profile. Crown produced in the midline, with eyes prominent. Forewing slightly to considerably narrowed apically and often with bright coloration; inner apical cell very short, second and fourth apical cells quadrilateral in shape, third apical cell triangular.

Pygofer usually with two well-differentiated lobes caudally, either one or both pigmented, serrated and bearing rigid microsetae terminally. Subgenital plate narrowed distally and hooked apically, except for a basal or near basal macroseta, several macrosetae near apical 1/3. Paramere with slightly varying proportions between caudal part and two remaining ones, usually with several sensory pores or fine microsetae on caudal part. Connective Y-shaped. Aedeagus with developed dorsoatrium; aedeagal shaft usually with two pairs of long apical appendages (*E.* (*S.*) *nasti* and *E.* (*S.*) *extensa* with both apical and lateral appendages).

### 3.2. Key to Species of Subgenus Singhardina from China (Males)

1.Pygofer always with hind margin lobed and upper lobe with a sclerotized protrusion on the dorsal margin; forewing covered by bright color patterns··············2

-Pygofer with hind margin not distinctly lobed or upper lobe prominently larger than lower one; forewing covered by brownish or brownish yellow patches···········38

2.Forewing with big light brownish to brownish black patches on RP vein··········································································································*E.* (*S.*) *rubra* group 3

-Forewing without patch or just with brownish short stripe on RP vein·······················12

3.Aedeagal shaft not extremely expanded or compressed··················································4

-Aedeagal shaft extremely expanded and compressed laterally or posteriorly··············6

4.Aedeagal shaft with a pair of apical appendages····························*E.* (*S.*) *galacta*
**sp. nov.**

-Aedeagal shaft with two pairs of apical appendages·····························································5

5.Ventral appendages of aedeagal shaft bifurcate apically········································································ *E.* (*S.*) *menglunensis* Huang et Zhang

-Ventral appendages of aedeagal shaft with a short branch at basal 1/3 view·····································································································*E.* (*S.*) *scamba*
**sp. nov.**

6.Aedeagal shaft not expanded from base············································································7

-Aedeagal shaft expanded from base···················································································8

7.Aedeagal shaft extremely expanded centrally···············*E.* (*S.*) *rubrocorona* Cai et Kuoh

-Aedeagal shaft extremely expanded apically·················*E.* (*S.*) *unipunctata* Hu et Kuoh

8.Aedeagal shaft compressed posteriorly·············································································9

-Aedeagal shaft compressed laterally················································································10

9.Aedeagal shaft with both dorsal and ventral appendages·····*E.* (*S.*) *foliiformis*
**sp. nov.**

-Aedeagal shaft with two pairs of dorsal appendages····················*E.* (*S.*) *scandens*
**sp. nov.**

10.Aedeagal shaft with ventral appendages bifurcate························*E.* (*S.*) *scalesa*
**sp. nov.**

-Aedeagal shaft with ventral appendages not bifurcate···················································11

11.Aedeagal shaft with ventral appendages parallel to dorsal ones in lateral view·····························································································*E.* (*S.*) *rubra* Dworakowska

-Aedeagal shaft with ventral and dorsal appendages overlapped basally in lateral view··········································································································*E.* (*S.*) *recta*
**sp. nov.**

12.Forewing with patches on basal part distinctly different from apical part; upper lobe of pygofer side with rigid microsetae distally; aedeagal shaft with apical appendages simple··································································································*E.* (*S.*) *vittata* group 13

-Forewing with patches on basal part not distinctly different from apical part; upper lobe of pygofer side serrated distally; aedeagal shaft with apical appendages variable·······················································································*E.* (*S.*) *punjabensis* group 14

13.Aedeagal shaft with ventral appendages not bifurcate··········*E.* (*S.*) *amacularis*
**sp. nov.**

-Aedeagal shaft with ventral appendages bifurcate··········*E.* (*S.*) *parilintanonica*
**sp. nov.**

14.Forewing white to yellowish with simple color patterns················································15

-Forewing bright with complex color patterns··································································20

15.Forewing with a large blackish patch on CuA″ vein························································16

-Forewing without large blackish patch on CuA″ vein····················································17

16.Aedeagal shaft with ventral appendages bifurcate apically and without short protrusion···············································································*E.* (*S.*) *flavicorona* Cai et Kuoh

-Aedeagal shaft with ventral appendages not bifurcate and with two short spines respectively on middle line and base····················· *E.* (*S.*) *diplopunctata* Huang et Zhang

17.Forewing without patches on center of clavus··················*E.* (*S.*) *zadyma* Dworakowska

-Forewing with patches on center of clavus·······································································18

18.Forewing with a big irregular patch on center of clavus······································································*E.* (*S.*) *unilobata* Chiang, Hsu et Knight

-Forewing with a half round patch at posterior margin of clavus··································19

19.Aedeagal shaft with ventral appendages not bifurcate············ *E.* (*S.*) *centralis* Yang et Li

-Aedeagal shaft with ventral appendages bifurcate·············· *E.* (*S.*) *cuii* Huang et Zhang

20.Forewing with distinctly bounded brown patches on center of clavus and adjoining corium, distal margin of brochosome area with broad oblique stripe···························21

-Forewing not as above·········································································································26

21.Aedeagal shaft with a pair of apical appendages···············································································*E.* (*S.*) *rubrivittata* Chiang, Hsu et Knight

-Aedeagal shaft with two or three pairs of apical appendages····················································22

22.Aedeagal shaft with three pairs of apical appendages·········· *E.* (*S.*) *exclamationis* Yang et Li

-Aedeagal shaft with two pairs of apical appendages···························································23

23.Dorsal appendages of aedeagal shaft bifurcate····························*E.* (*S.*) *biavis* Yang et Li

-Dorsal appendages of aedeagal shaft not bifurcate·························································24

24.Ventral appendages of aedeagal shaft with a short branch near basal 1/3···············································································································*E.* (*S.*) *lata*
**sp. nov.**

-Ventral appendages of aedeagal shaft bifurcate or trifurcate·········································25

25.Ventral appendages of aedeagal shaft trifurcate····················· *E.* (*S.*) *wuyiana* Yang et Li

-Ventral appendages of aedeagal shaft bifurcate···················*E.* (*S.*) *dazhulana* Yang et Li

26.Crown and pronotum without patch················································································27

-Crown and pronotum with patches···················································································28

27.Aedeagal shaft with three pairs of apical appendages··*E.* (*S.*) *tripunctata* Huang et Zhang

-Aedeagal shaft with two pairs of apical appendages···················· *E.* (*S.*) *uprotrusa*
**sp. nov.**

28.Aedeagal shaft with a pair of apical appendages············*E.* (*S.*) *flaviscutella*
**sp. nov.**

-Aedeagal shaft with two or more than two pairs of apical appendages·································29

29.Aedeagal shaft with three or four pairs of appendages·······························································30

-Aedeagal shaft with two pairs of apical appendages···························································32

30.Aedeagal shaft with four pairs of apical appendages·················*E.* (*S.*) *gracilifurca*
**sp. nov.**

-Aedeagal shaft with three pairs of apical appendages and a pair of lateral appendages···························································································································31

31.Ventral appendages of aedeagal shaft bifurcate····················· *E.* (*S*) *nasti* Dworakowska

-Ventral appendages of aedeagal shaft not bifurcate····················· *E.* (*S.*) *extensa*
**sp. nov.**

32.Ventral appendages of aedeagal shaft serrated basally·····*E.* (*S.*) *spinifera* Huang et Zhang

-Ventral appendages of aedeagal shaft not serrated basally············································33

33.Ventral appendages of aedeagal shaft with four branches···*E.* (*S.*) *fasciata* Dworakowska

-Ventral appendages of aedeagal shaft bifurcate······························································34

34.Ventral appendages of aedeagal shaft with upper branch thinner than lower ones······················································································*E.* (*S.*) *rubrania* Huang et Zhang

-Ventral appendages of aedeagal shaft with two branches subequal in width·················35

35.Dorsal appendages of aedeagal shaft bifurcate·················· *E.* (*S.*) *choui* Huang et Zhang

-Dorsal appendages of aedeagal shaft not bifurcate·························································36

36.Ventral appendages of aedeagal shaft bifurcate apically········ *E.* (*S.*) *anurous* Zhang et Xiao

-Ventral appendages of aedeagal shaft bifurcate centrally··············································37

37.Ventral appendages of aedeagal shaft with outer branch and dorsal appendages subequal in length······································································*E.* (*S.*) *rutilans* Hu et Kuoh

-Ventral appendages of aedeagal shaft with outer branch longer than dorsal appendages·····················································································*E.* (*S.*) *fusca* Cai et Kuoh

38.Forewing with big brownish patches on clavus and adjacent areas of corium; subgenital plate with narrowed apex not distinctly extended and with a row of moderate or short macrosetae extending from apical 1/3 to subapex; paramere with caudal part distinctly shorter than combined length of anterior and central parts; aedeagal shaft with dorsal appendages not bifurcate and ventral ones bifurcate····························································································*E.* (*S.*) *mamata* group 39

-Forewing with various brown to yellowish brown shades; subgenital plate narrowed distally and hooked apically, except for a basal or near basal macroseta, two or more moderate macrosetae longitudinally near apical 1/3; paramere with caudal part subequal to combined length of anterior and central parts; aedeagal shaft often narrow with two pairs of unbranched apical appendages··············*E.* (*S.*) *robusta* group 44

39.Ventral appendages of aedeagal shaft with lower branch indistinct····················································································*E.* (*S.*) *prima* Dworakowska

-Ventral appendages of aedeagal shaft with lower branch distinct·······························40

40.Ventral appendages of aedeagal shaft with upper branch shorter than lower one··························································································*E.* (*S.*) *acapitata* Dworakowska

-Ventral appendages of aedeagal shaft with upper branch longer than lower one······41

41.Ventral appendages of aedeagal shaft with lower branch short and not reaching 1/3 upper ones·····························································································································42

-Ventral appendages of aedeagal shaft with lower branch long and reaching 1/2 upper ones········································································································································43

42.Forewing with longitudinally S-shaped markings··············*E.* (*S.*) *flavistriata* Yang et Li

-Forewing with interrupted markings····················*E.* (*S.*) *yingfengica* Dworakowska

43.Pronotum with four round patches posteriorly·······················*E.* (*S.*) *jarrayi* Dworakowska

-Pronotum with two round patches anteriorly·····················*E.* (*S.*) *quadrimacularis*
**sp. nov.**

44.Aedeagal shaft just with dorsal appendages··············*E.* (*S.*) *immatura* Zhang et Huang

-Aedeagal shaft with both ventral and dorsal appendages··········································45

45.Dorsal appendages of aedeagal shaft not bifurcate······················································46

-Dorsal appendages of aedeagal shaft bifurcate···············*E.* (*S.*) *furca* Zhang et Huang

46.Ventral appendages of aedeagal shaft bifurcate·······*E.* (*S.*) *dissimilis* Zhang et Huang

-Ventral appendages of aedeagal shaft not bifurcate·····················································47

47.Dorsal appendages of aedeagal shaft with a short protrusion··················································································*E.* (*S.*) *flatilis* Zhang et Huang

-Dorsal appendages of aedeagal shaft without protrusion············································48

48.Aedeagal shaft with ventral appendages circularly bent in lateral view··················································································· *E.* (*S.*) *pookiewica* Dworakowska

-Aedeagal shaft with ventral appendages slightly curved in lateral view··················49

49.Aedeagal shaft with dorsal and ventral appendages subequal in width····················································································*E.* (*S.*) *fumosa* Zhang et Huang

-Aedeagal shaft with dorsal appendages thinner than ventral ones·····················································································*E.* (*S.*) *krispinilla* Dworakowska

### 3.3. Species Descriptions

#### 3.3.1. The *Eurhadina* (*Singhardina*) *rubra* Species Group from China

*E.* (*S.*) *foliiformis*
**sp. nov.**, *E.* (*S.*) *galacta*
**sp. nov.**, *E.* (*S.*) *menglunensis*, *E.* (*S.*) *recta*
**sp. nov.**, *E.* (*S.*) *rubra*, *E.* (*S.*) *rubrocorona*, *E.* (*S.*) *scalesa*
**sp. nov.**, *E.* (*S.*) *scamba*
**sp. nov.**, *E.* (*S.*) *scandens*
**sp. nov.**, and *E.* (*S.*) *unipunctata*.

**Description.** Coloration bright. Forewing with or without pink patch (*E.* (*S.*) *galacta* and *E.* (*S.*) *scandens* pale white); distal margin of brochosome area and ScP + RA vein with a brownish black oblique stripe; RP vein with a big brownish-black patch.

Pygofer often with two well-differentiated lobes caudally; upper lobe with several rigid microsetae terminally (*E.* (*S.*) *galacta* with a sclerotized protrusion dorsally); lower lobe often involuted with finger-like appendage distally (*E.* (*S.*) *foliiformis*, *E.* (*S.*) *galacta* and *E.* (*S.*) *unipunctata* without finger-like appendage). Subgenital plate triangular with row of numerous differentiated moderate or short macrosetae; several fine microsetae near subapex and a few rigid microsetae on apex. Paramere with caudal part equal to or shorter than combined length of anterior and central parts, usually with a row of fine microsetae on outer margin and sensory pores extending from inner margin to outer margin on caudal part. Connective slender with central ridge long. Aedeagal shaft extremely expanded from base and compressed laterally or posteriorly (*E.* (*S.*) *galacta*, *E.* (*S.*) *menglunensis* and *E.* (*S.*) *scamba* not expanded or compressed) with apical appendages complicated in shape.

##### *Eurhadina* (*Singhardina*) *foliiformis* Zhang et Huang **sp. nov.**

(Figure 1A–M)

Body form and color patterns very similar to *E.* (*S.*) *rubra*, but brochosome area with a broad oblique stripe along distal margin.

**Male genitalia**: Lobe of pygofer side rounded, lower lobe not involuted and without finger-like appendage distally. Subgenital plate with long macroseta near basal 1/4; four moderate and four short macrosetae longitudinally from apical 1/3 to subapex and three rigid microsetae on apex. Connective with lateral arms weakly separated. Paramere with caudal part equal to combined length of anterior and central parts, five fine microsetae and sensory pores on caudal part. Aedeagal shaft slightly expanded from base and compressed posteriorly with two pairs of leaf-shaped and unbranched apical appendages; dorsal appendages slightly wider than ventral one.

**Measurement.** Length of male 3.38–3.54 mm (including wings).

**Material examined.** Holotype: ♂, China, Yunnan Prov., Sanchahe, 1991.VI.7, coll. Rungang Tian. Paratypes: 22 ♂, same data with holotype.

**Etymology.** The specific epithet is derived from the Latin word “*foliiformis*”, referring to aedeagal shaft with apical appendages leaf-shaped.

**Notes.** This new species resembles *E.* (*S.*) *rubra* in external characteristics, but differs in brochosome area with a wider oblique stripe along distal margin and the aedeagal shaft having two pairs of wide and leaf-like apical appendages.

##### *Eurhadina* (*Singhardina*) *galacta* Zhang et Huang **sp. nov.**

(Figure 2A–M)

Body creamy. Face sordid white. Crown, pronotum, and scutellum without patch. Scutellum yellowish pale, basal triangles yellowish green. Forewing pale white, without patch on basal half of corium and clavus; distal margin of brochosome area, ScP + RA vein, and lateral margin of apical area with light brownish oblique stripes; RP vein with a big light brownish patch centrally; first apical cell and its adjoining second apical cell covered with a semicircular yellowish-brown patch.

**Male genitalia**: Upper lobe of pygofer side rounded with a sclerotized protrusion dorsally. Subgenital plate slender with long macroseta near basal 1/4; three moderate and four short macrosetae longitudinally from apical 1/2 to subapex; several rigid microsetae on subapex. Connective long with lateral arms weakly separated. Paramere slightly expanded apically with caudal part equal to combined length of anterior and central parts, three fine microsetae and five sensory pores on caudal part. Aedeagal shaft with a pair of dorsal appendages branched basally; longer appendages slightly expanded centrally, arched toward outside and more than twice as long as inner branches; inner branches draw nearer to each other apically.

**Measurement.** Length of male 3.71–3.95 mm (including wings).

**Material examined.** Holotype: ♂, China, Yunnan Prov., Mt. Weibao, 2001.VII.20, coll. Qiang Sun. Paratypes: 4 ♂, same data with holotype.

**Etymology.** This specific epithet is derived from the Greek word “*galactos*”, referring to milky white forewing.

**Notes.** This new species resembles *E.* (*S.*) *scandens*
**sp. nov.** in external characteristics and male genitalia, but it differs by the aedeagal shaft having a pair of dorsal appendages branched at base.

##### *Eurhadina* (*Singhardina*) *menglunensis* Huang et Zhang, 1999

(Figure 3A–M)

*Eurhadina* (*Singhardina*) *menglunensis* Huang et Zhang, 1999: 252, Figure 6 [8].

**Material examined.** 1 ♂ (holotype), China, Yunnan Prov., Menglun, 1982.IV.10, coll.

Jingruo Zhou and Sumei Wang.

**Distribution.** China (Yunnan).

##### *Eurhadina* (*Singhardina*) *recta* Zhang et Huang **sp. nov.**

(Figure 4A–M)

Body form and color patterns very similar to *E.* (*S.*) *scalesa*
**sp. nov.**, but somewhat longer and face with lighter coloration.

**Male genitalia**: Lobe of pygofer side brownish black with upper lobe serrated and lower lobe strongly pigmented. Subgenital plate with long macroseta near basal 1/3, seven moderate macrosetae longitudinally near apical 1/3 to subapex; three rigid microsetae near subapex and four on apex. Connective long with lateral arms widely separated. Paramere with caudal part shorter than combined length of anterior and central parts, seven fine microsetae and five sensory pores on caudal part. Aedeagal shaft with two pairs of unbranched apical appendages; dorsal appendages slightly expanded centrally and reaching 2/3 ventral ones; ventral appendages slender and simple.

**Measurement.** Length of male 3.46–3.79 mm (including wings).

**Material examined.** Holotype: ♂, China, Fujian Prov., Sangang, 1984.VIII.8, coll. Zhixin Cui. Paratypes: 8 ♂, same data with holotype; 5 ♂, China, Hunan Prov., Mt. Mang, 1985.VII.31, 2 ♂, 1985.VII.30, 2 ♂, 1985.VII.29, 2 ♂, 1985.VII.27, coll. Yalin Zhang and Yonghui Cai; 1 ♂, China, Guangxi Prov., Jinxiu, 1983.V.27, coll. Sikong Liu; 1 ♂, China, Guangxi Prov., Mt. Dayao, 1982.VI.13, coll. Jikun Yang; 50 ♂, China, Guangdong Prov., Chebaling, 2020.VII.17, coll. Junjie Wang.

**Etymology.** The specific epithet is derived from the Latin word “*rectus*”, referring to dorsal appendages of aedeagal shaft bended at right angle in posterior view.

**Notes.** This new species resembles *E.* (*S.*) *rubra* in external characteristics and male genitalia, but it can be distinguished from the latter by aedeagal shaft with dorsal appendages shorter than ventral ones.

##### *Eurhadina* (*Singhardina*) *rubra* Dworakowska, 1969

(Figure 5A–L)

*Eurhadina* (*Singhardina*) *rubra* Dworakowska, 1969: 76, Figures 22, 32, 33, 40, 47, 52, 57, 91, 96, 105 [2]; Zhang, 1990: 149, Figure 165 [5]; Dworakowska, 2002: 58, Figures 92–100 [3].

**Material examined.** 23 ♂, China, Yunnan Prov., Sanchahe, 1991.VI.7, coll. Rungang Tian; 8 ♂, Sangang, China, Fujian Prov., Mt. Wuyi, 2003.VI.18, coll. Yani Duan.

**Distribution.** China (Fujian, Hunan, Yunnan); Malaysia; Vietnam.

##### *Eurhadina* (*Singhardina*) *rubrocorona* Cai et Kuoh, 1993

*Eurhadina* (*Singhardina*) *rubrocorona* Cai et Kuoh, 1993: 223, Figure 2 [9].

**Material examined.** 8 ♂, China, Fujian Prov., Sangang, 1984.VIII.8, coll. Zhixin Cui; 1 ♂, China, Fujian Prov., Sangang, 1960.VI.25, coll. Yiran Zhang; 1 ♂, China, Guizhou Prov., Mt. Fanjing, 2001.VII.28, coll. Qiang Sun; 3 ♂, China, Zhejiang Prov., Mt. Fengyang, 2003.VIII.08, coll. Wu Dai; 1 ♂, China, Zhejiang Prov., Mt. Gutian, 2003.VIII.17, coll. Wu Dai; 1 ♂, China, Zhejiang Prov., Mt. Tianmu, 2003.VIII.28, coll. Wu Dai; 3 ♂, China, Zhejiang Prov., Wuyanling, 2005.VII.28, coll. Yani Duan; 38 ♂, China, Fujian Prov., Mt. Wuyi, 2008.VIII.17, coll. Xia Gao and Xiaoting Li.

**Distribution.** China (Zhejiang, Fujian, Guizhou).

##### *Eurhadina* (*Singhardina*) *scalesa* Zhang et Huang **sp. nov.**

(Figure 6A–M)

Body form and color patterns very similar to *E.* (*S.*) *rubra*, but face brownish black and somewhat shorter.

**Male genitalia**: Lobe of pygofer side pigmented terminally, upper lobe rounded distally. Subgenital plate with long macroseta near basal 1/3; seven moderate and one short macrosetae longitudinally near apical 1/3 to subapex; three rigid microsetae on apex. Connective long with lateral arms weakly separated. Paramere with caudal part shorter than combined length of anterior and central parts, six fine microsetae and four sensory pores on caudal part. Aedeagal shaft extremely expanded from base and compressed laterally with two pairs of slender apical appendages; dorsal appendages unbranched; ventral appendages bifurcate with two branches nearly 90° angled in posterior view and outer branches equal to 1/2 inner ones in length.

**Measurement.** Length of male 2.91–3.11 mm (including wing).

**Material examined.** Holotype: ♂, China, Yunnan Prov., Sanchahe, 1991.VI.7, coll. Rungang Tian. Paratypes: 21 ♂, same data with holotype; 9 ♂, China, Yunnan Prov., Bubang, 2010.Ⅸ.4, coll. Meng Zhang.

**Etymology.** The specific epithet is the Latinized English word “*scalesa*”, referring to the ventral appendages of aedeagal shaft resembling a balance or scale in posterior view.

**Notes.** This new species resembles *E.* (*S.*) *rubra* in external characters and genitalia, but it differs by the ventral appendages of aedeagal shaft bifurcate with two branches nearly 90° angled in posterior view.

##### *Eurhadina* (*Singhardina*) *scamba* Zhang et Huang **sp. nov.**

(Figure 7A–L)

Body yellowish brown. Face brownish. Crown, pronotum and scutellum without patch. Scutellum yellowish and basal triangles pale yellowish green. Forewing pale white with clavus and its adjacent part of corium light yellowish brown; central part of corium pinkish; distal margin of brochosome area and ScP + RA vein with slender oblique stripe.

**Male genitalia**: Both lobes of pygofer side pigmented terminally, dorsal margin of lower lobe with several rigid microsetae laterally. Subgenital plate with long macroseta near basal 1/4, six short macrosetae near apical 1/3; several rigid microsetae near subapex and two on apex. Connective long with lateral arms weakly separated. Paramere with caudal part shorter than combined length of anterior and central parts, six fine microsetae and sensory pores on caudal part. Aedeagal shaft slim and arched in lateral view with two pairs of apical appendages; ventral appendages long and wide with a short branch at basal 1/3 arched toward outside; dorsal ones slender and simple.

**Measurement.** Length of male 2.96–3.16 mm (including wings).

**Material examined.** Holotype: ♂, China, Yunnan Prov., Sanchahe, 1991.VI.7, coll. Rungang Tian. Paratypes: 25 ♂, same data with holotype.

**Etymology.** The specific epithet is derived from the Greek word “*scambos*”, referring to aedeagal shaft with apical appendages curved toward outside in dorsal view.

**Notes.** This new species resembles *E.* (*S.*) *rubra* in external characteristics and male pygofer side lobe, but it differs from the latter in having a slim aedeagal shaft and dorsal appendages of aedeagal shaft with a branch basally.

##### *Eurhadina* (*Singhardina*) *scandens* Zhang et Huang **sp. nov.**

(Figure 8A–M)

Body creamy. Face brownish. Crown, pronotum and scutellum without patch. Forewing pale white, without patch on basal half of corium and clavus; corium pinkish centrally; distal margin of brochosome area, ScP + RA vein and lateral margin of apical area with slender light brownish oblique stripes, RP vein with a big light brownish patch.

**Male genitalia**: Lobe of pygofer side strongly pigmented terminally and upper lobe rounded. Subgenital plate with long macroseta near basal 1/4, three moderate and four short macrosetae longitudinally from apical 1/3 to subapex; several rigid microsetae near apical 1/3 and two on apex. Connective long with lateral arms weakly separated. Paramere blunt terminally with caudal part shorter than combined length of anterior and central parts, five fine microsetae and four sensory pores on caudal part. Aedeagal shaft slightly expanded from base and compressed posteriorly with two pairs of slender and simple dorsal appendages merged basally; lower dorsal appendages equal to 1/3 upper ones in length and four appendages arched toward four different directions in posterior view.

**Measurement.** Length of male 3.33–3.53 mm (including wings).

**Material examined.** Holotype: ♂, China, Yunnan Prov., Sanchahe, 1991.VI.7, coll. Rungang Tian. Paratypes: 22 ♂, same data as holotype.

**Etymology.** The specific epithet is derived from the Latin word “*scando*”, referring to four apical appendages of aedeagal shaft seeming to climb in posterior view.

**Notes.** This new species resembles *E.* (*S.*) *scamba*
**sp. nov.** in the male genitalia, but differs from the latter in pale white forewing and the aedeagal shaft with unbranched apical appendages pointing toward four different directions in posterior view.

##### *Eurhadina* (*Singhardina*) *unipunctata* Hu et Kuoh, 1991

*Eurhadina* (*Singhardina*) *unipunctata* Hu et Kuoh, 1991: 259, Figure 5 [10].

**Distribution.** China (Yunnan).

#### 3.3.2. The *Eurhadina* (*Singhardina*) *vittata* Species Group from China

*E.* (*S.*) *amacularis*
**sp. nov.** and *E.* (*S.*) *parilintanonica*
**sp. nov.**

**Description.** Coloration involves contrasting blackish patches and usually other bright colors. Forewing with patches on basal part distinctly different from apical part; distal margin of brochosome area without oblique stripe; RP vein without big brownish-black patch.

Pygofer rounded; when upper lobe narrowed, usually with prominent short protrusion dorsally (without protrusion in *E.* (*S.*) *parilintanonica*) and with several rigid microsetae terminally; when upper lobe not narrowed and large, protrusion obliterated; lower lobe usually not serrated (serrated in *E.* (*S.*) *amacularis*). Paramere blunt terminally with a row of tightly fine microsetae and sensory pores near outer margin on caudal part. In relation to sternite 9, subgenital plate short with a row of short or moderate macrosetae and fine microsetae on apical 1/3, several rigid microsetae near subapex. Connective long. Aedeagal shaft with two pairs of apical appendages, dorsal appendages not bifurcate, ventral ones often bifurcate.

##### *Eurhadina* (*Singhardina*) *amacularis* Zhang et Huang **sp. nov.**

(Figure 9A–L)

Body creamy. Crown, pronotum, and face yellowish orange. Crown with three creamy patches, remainder orange. Pronotum with two pairs of symmetrically oval creamy patches centrally and several small and irregular creamy patches anteriorly, remainder orange; basal triangles orange; scutellum yellowish pale with brownish apex, an orange patch centrally and two blackish marks on sides of scutellum. Forewing transparent with reddish patches on basal part of corium and clavus; basal margin of brochosome area yellow.

**Male genitalia**: Upper lobe of pygofer side large, lower lobe serrated and slightly pigmented distally. Subgenital plate with long macroseta near basal 1/3, six short macrosetae extend from apical 1/3 to subapex and several rigid microsetae on subapex. Connective with lateral arms widely separated. Paramere broadened with three fine microsetae and few sensory pores on caudal part. Aedeagal shaft slim and curved, with two pairs of unbranched apical appendages; ventral appendages sinuate and shorter than dorsal ones, dorsal appendages widened centrally.

**Measurement.** Length of male 3.15–3.34 mm (including wings).

**Material examined.** Holotype: ♂, China, Yunnan Prov., Sanchahe, 1991.VI.7, coll. Rungang Tian. Paratypes: 21 ♂, same data as holotype.

**Etymology.** The specific epithet is derived from the Latin words “*a*” and “*macula*” referring to forewing lacking a patch except basal part of corium and clavus.

**Notes.** This new species belongs to the *E.* (*S.*) *vittata* group based on male genitalia, but differs from other species of this group in forewing transparent without round black patch on basal part of clavus and dorsal appendages of aedeagal shaft unbranched.

##### *Eurhadina* (*Singhardina*) *parilintanonica* Zhang et Huang **sp. nov.**

(Figure 10A–M)

Body transparent, face sordid white. Crown creamy with four small orange patches anteriorly. Pronotum with orange fascia anteriorly and two blackish round patches posteriorly. Scutellum yellowish orange, basal triangles blackish. Forewing transparent without patch on basal half of corium; clavus with a blackish round patch centrally and brochosome area with a short traverse stripe on costal margin.

**Male genitalia**: Upper lobe of pygofer side tuberculated distally, narrowed but without prominent short protrusion dorsally. Subgenital plate with long macroseta near basal 1/5, four short macrosetae near apical 1/3; two rigid microsetae on apex. Paramere slender and straight with three fine microsetae and four sensory pores on caudal part. Aedeagal shaft slim and curved, with two pairs of apical appendages; ventral appendages long and not branched, located between two branches of dorsal appendages; dorsal appendages bifurcate with inner branched with semi-lunar depression on apical 1/2 and wider than outer ones.

**Measurement.** Length of male 3.34–3.62 mm (including wings).

**Material examined.** Holotype: ♂, China, Yunnan Prov., Mengyang, 1991.VI.7, coll. Wanzhi Cai and Yinglun Wang. Paratypes: 17 ♂, China, Yunnan Prov., Dianlongmen, 2009.V.15, coll. Wei Cui.

**Etymology.** The specific epithet is derived from the combination of the Latin word “*pa**rilis*” and the name of the related species “*intanonica*”, referring to this new species closely resembling *E.* (*S.*) *intanonica*.

**Notes.** This new species resembles *E.* (*S.*) *intanonica* in external characteristics and male genitalia, but differs from the latter in aedeagal shaft with ventral appendages U-shaped and appearing between the two branches of dorsal appendages in dorsal view.

#### 3.3.3. The *Eurhadina* (*Singhardina*) *punjabensis* Species Group from China

*E.* (*S.*) *anurous*, *E.* (*S.*) *biavis*, *E.* (*S.*) *centralis*, *E.* (*S.*) *choui*, *E.* (*S.*) *cuii*, *E.* (*S.*) *dazhulana*, *E.* (*S.*) *diplopunctata*, *E.* (*S.*) *exclamationis*, *E.* (*S.*) *extensa*
**sp. nov.**, *E.* (*S.*) *fasciata*
**rec. nov.**, *E.* (*S.*) *flavicorona*, *E.* (*S.*) *flaviscutella*
**sp. nov.**, *E.* (*S.*) *fusca*, *E.* (*S.*) *gracilifurca*
**sp. nov.**, *E.* (*S.*) *lata*
**sp. nov.**, *E.* (*S.*) *nasti*, *E.* (*S.*) *rubrania*, *E.* (*S.*) *rubrivittata*, *E.* (*S.*) *rutilans*, *E.* (*S.*) *spinifera*, *E.* (*S.*) *tripunctata*, *E.* (*S.*) *unilobata*, *E.* (*S.*) *uprotrusa*
**sp. nov.**, *E.* (*S.*) *wuyiana* and *E.* (*S.*) *zadyma*
**rec. nov.**

**Description.** Coloration bright. Forewing always with various yellowish, orange to reddish coloration; clavus with or without brownish patch; distal margin of brochosome area with oblique stripe; RP vein without big brownish-black patch.

Pygofer often with two well-differentiated lobes caudally; upper lobe usually with a sclerotized protrusion on the dorsal margin close to dorso-caudal angle and with several rigid microsetae terminally; lower lobe pigmented. Subgenital plate triangular with numerous differentiated macrosetae and fine microsetae apically. Connective slender. Paramere with caudal part subequal to or slightly shorter than combined length of anterior and central parts, broadened subapically and bluntly terminated with numerous differentiated fine microsetae and sensory pores on caudal part. Aedeagal shaft often expanded with two pairs of apical appendages, at least one pair of appendages bifurcate.

##### *Eurhadina* (*Singhardina*) *anurous* Zhang et Xiao, 2000

*Eurhadina* (*Singhardina*) *anurous* Zhang et Xiao, 2000: 110, Figures 15–26 [11].

**Distribution.** China (Yunnan).

##### *Eurhadina* (*Singhardina*) *biavis* Yang et Li, 1991

(Figure 11A–H)

*Eurhadina* (*Singhardina*) *biavis* Yang et Li, 1991: 25, Figures 13–18 [12].

*Eurhadina rubromia* Cai et Kuoh, 1993: 225, Figure 4 [9], **syn. nov.**

**Material examined.** 1 ♂ (holotype), China, Fujian Prov., Chongan, Sangang, 1979.VI.27, coll. Jikun Yang; 3 ♂, China, Fujian Prov., Sangang, 1979.VI.27, coll. Jikun Yang; 10 ♂, China, Fujian Prov., Sangang, 1984.VIII.8, coll. Zhixin Cui; 1 ♂, China, Yunnan Prov., Tengchong, Dahaoping, 1999.XI.25, coll. Irena Dworakowska; 1 ♂, China, Yunnan Prov., Tengchong, Dahaoping, 1999.XI.27, coll. Irena Dworakowska; 3 ♂, China, Fujian Prov., Mt. Fengyang, 2003.VIII.08, coll. Wu Dai; 1 ♂, China, Zhejiang Prov., Mt. Tianmu, 2003.VIII, coll. Wu Dai; 2 ♂, China, Zhejiang Prov., Mt. Gutian, 2003.VIII.17, coll. Wu Dai; 4 ♂, China, Zhejiang Prov., Wuyanling, 2005.VII.28, coll. Yani Duan; 3 ♂, China, Zhejiang Prov., Wuyanling, 2005.VII.30, coll. Yani Duan; 1 ♂, China, Zhejiang Prov., Wuyanling, 2005.VIII.01, coll. Yani Duan; 3 ♂, China, Fujian Prov., Mt. Wuyi, 2008.VIII.16, coll. Bin Xiao; 2 ♂, China, Fujian Prov., Mt. Wuyi, 2008.VIII.16, coll. Xia Gao and Xiaoting Li; 10 ♂, China, Fujian Prov., Mt. Wuyi, 2008.VIII.17, coll. Xia Gao and Xiaoting Li; 1 ♂, China, Fujian Prov., Mt. Wuyi, 2008.VIII.17, coll. Manqiang Wang.

**Distribution.** China (Fujian, Yunnan).

**Notes.** We compared the habitus, detected genitalia, and illustrated figures of the holotype specimen of *E.* (*S.*) *biavis* Yang et Li [12] with *Eurhadina rubromia* as described by Cai et Kuoh in 1993 [9], and found their localities are all in Fujian and possess identical characteristics: yellowish orange body, scutellum with apex blackish, central clavus with a big black irregular patch like a waterfowl, the aedeagus with three pairs of apical appendages, dorsal appendages of aedeagal shaft bi-forked apically, and ventral ones branched basally. We therefore consider them to be the same species and, according to the rule of ICZN, we propose *Eurhadina rubromia* Cai et Kuoh as a junior synonymy to *E.* (*S.*) *biavis* Yang et Li.

##### *Eurhadina* (*Singhardina*) *centralis* Yang et Li, 1991

(Figure 12A–M)

*Eurhadina* (*Singhardina*) *centralis* Yang et Li, 1991: 26, Figures 25–31 [12].

**Material examined.** 1 ♂, China, Fujian Prov., Dehua, 1974.XI.12, coll. Jikun Yang; 2 ♀, China, Hunan Prov., Mt. Mang, 1985.VII.31, coll. Yalin Zhang and Yonghui Cai.

**Distribution.** China (Fujian, Zhejiang, Hunan).

##### *Eurhadina* (*Singhardina*) *choui* Huang et Zhang, 1999

*Eurhadina* (*Singhardina*) *choui* Huang et Zhang, 1999: 247, Figure 2 [8].

**Material examined.** 2 ♂, China, Fujian Prov., Mt. Wuyi, 2008.VIII.17, coll. Xia Gao and Xiaoting Li; 1 ♂, China, Fujian Prov., Mt. Wuyi, 2008.VIII.16, coll. Xia Gao and Xiaoting Li; 1 ♂, China, Fujian Prov., Mt. Fengyang, 2008.VIII.01, coll. Xia Gao and Xiaoting Li.

**Distribution.** China (Fujian, Hunan).

##### *Eurhadina* (*Singhardina*) *cuii* Huang et Zhang, 1999

*Eurhadina* (*Singhardina*) *cuii* Huang et Zhang, 1999: 251, Figure 5 [8]; Dworakowska, 2002: 62 [3].

**Distribution.** China (Fujian).

##### *Eurhadina* (*Singhardina*) *dazhulana* Yang et Li, 1991

*Eurhadina* (*Singhardina*) *dazhulana* Yang et Li, 1991: 23, Figures 1–6 [12].

**Material examined.** 1 ♂, China, Fujian Prov., Jianyang, Dazhulan, 1974.X.27, coll. Fasheng Li; 1 ♀, China, Yunnan Prov., Mengla, Mt. Nangong, 1999.XII.17, coll. Irena Dworakowska; 2 ♀, China, Hunan Prov., Mt. Mang, 1985.VII.13, coll. Yalin Zhang and Yonghui Cai; 3 ♂, China, Fujian Prov., Mt. Wuyi, 2008.VIII.17, coll. Xia Gao and Xiaoting Li; 1 ♂, China, Yunnan Prov., Yaoqu, 2009.Ⅵ.3, coll. Wei Cui; 1 ♂, China, Zhejiang Province, Wulingkeng, 2003.VIII, coll. Wu Dai; 4 ♂, China, Zhejiang Prov., Wuyanling, 2005.VII.28, coll. Yani Duan; 1 ♂, China, Zhejiang Prov., Wuyanling, 2005.VII.29, coll. Yani Duan; 1 ♂, China, Zhejiang Prov., Wuyanling, 2005.VIII.02, coll. Yani Duan.

**Distribution.** China (Zhejiang, Fujian, Hunan, Yunnan).

##### *Eurhadina* (*Singhardina*) *diplopunctata* Huang et Zhang, 1999

(Figure 13A–K)

*Eurhadina* (*Singhardina*) *diplopunctata* Huang et Zhang, 1999: 253, Figure 7 [8].

**Material examined.** 26 ♂ (Paratypes), China, Fujian Prov., Sangang, 1984.VIII.8, coll. Zhixin Cui; 5 ♂, China, Fujian Prov., Sangang, 1984.VII.17, coll. Zhixin Cui; 1 ♂, China, Hunan Prov., Mt. Mang, 1985.VII.31, coll. Yalin Zhang and Yonghui Cai; 3 ♂, China, Hunan Prov., Mt. Mang, 1984.VII.9, coll. Zhixin Cui; 7 ♂, China, Zhejiang Prov., Wuyanling, 2005.VII.28, coll. Yani Duan; 3 ♂, China, Zhejiang Prov., Wuyanling, 2005.VIII.02, coll. Yani Duan; 4 ♂, China, Zhejiang Prov., Wuyanling, 2005.VII.30, coll. Yani Duan; 8 ♂, China, Zhejiang Prov., Mt. Gutian, 2003.VIII.17, coll. Wu Dai; 2 ♂, China, Zhejiang Prov., Wuyanling, 2007.VIII.07, coll. Xinmin Zhang; 2 ♂, China, Yunnan Prov., Yaoqu, 2009.VI.03, coll. Wei Cui; 7 ♂, China, Fujian Prov., Mt. Wuyi, 2008.VIII.16, coll. Xia Gao and Xiaoting Li; 21 ♂, China, Fujian Prov., Mt. Wuyi, 2008.VIII.17, coll. Xia Gao and Xiaoting Li.

**Distribution.** China (Fujian, Zhejiang).

##### *Eurhadina* (*Singhardina*) *exclamationis* Yang et Li, 1991

*Eurhadina* (*Singhardina*) *exclamationis* Yang et Li, 1991: 23, Figures 7–12 [12]; Dworakowska, 2002: 62 [3].

**Material examined.** 2 ♂, China, Fujian Prov., Sangang, 1979.VI.27, coll. Jikun Yang; 3 ♂, China, Fujian Prov., Sangang, 1984.VIII.8, coll. Zhixin Cui; 1 ♂, China, Fujian Prov., Sangang, 1960.VI.25, coll. Yiran Zhang; 5 ♂, China, Fujian Prov., Mt. Wuyi, 2008.VIII.17, coll. Xia Gao and Xiaoting Li.

**Distribution.** China (Fujian).

##### *Eurhadina* (*Singhardina*) *extensa* Zhang et Huang **sp. nov.**

(Figure 14A–M)

Body brownish pale. Face brownish. Crown creamy with a triangle brownish patch anteriorly and two orange patches posteriorly. Pronotum creamy with a semi-cycle brownish patch posteriorly and six brownish-black spots around it symmetrically; basal triangles brownish; scutellum yellowish with apex blackish. Forewing brownish pale, color pattern shown in Figures 170 and 171.

**Male genitalia:** Lobes of pygofer side long; upper lobe with a sclerotized protrusion terminally, lower lobe not pigmented and disc with two rigid setae. Subgenital slender, with long macroseta near basal 1/3; five short macrosetae extend from apical 1/2 to subapex; three rigid microsetae on apex. Connective long and with lateral arms weakly separated. Paramere with four sensory pores and five fine microsetae on caudal part. Aedeagal shaft expanded basally and compressed laterally with three pairs of unbranched apical appendages; ventral appendages V-shaped; dorsal appendages expand centrally, S-shaped and pointed outside in dorsal view; lateral appendages shorter than others, slender and slightly arched toward each other.

**Measurement.** Length of male 2.98–3.17 mm (including wings).

**Material examined.** Holotype: ♂, China, Yunnan Prov., Sanchahe, 1991.VI.7, coll. Rungang Tian. Paratypes: 8 ♂, same data as holotype.

**Etymology.** The specific epithet is derived from the Latin word “*extensivus*”, referring to aedeagal shaft extremely expanded basally.

**Notes.** This new species resembles *E.* (*S.*) *choui* in color pattern of forewing, but it can differ from the latter by aedeagal shaft expanded basally and with a pair of lateral appendages.

##### *Eurhadina* (*Singhardina*) *fasciata* Dworakowska, 2002, **rec. nov.**

(Figure 15A–L)

*Eurhadina* (*Singhardina*) *fasciata* Dworakowska, 2002: 56, Figures 73–82 [3].

**Material examined.** 1 ♂, China, Yunnan Prov., Mengyang, 1991.VI.7, coll. Yinglun Wang and Rungang Tian; 2 ♂, China, Yunnan Prov., Mengyang, 1991.VI.7, coll. Wanzhi Cai and Yinglun Wang; 1 ♂, China, Yunnan Prov., Mengyang, 1991.VI.7, coll. Guangchun Liu and Wanzhi Cai; 2 ♂ Wei Cui; China, Jiangxi Prov., Bayingxiang, 2004.Ⅷ.15, coll. Cong Wei and Meixia Yang; 2 ♂ Wei Cui; China, Jiangxi Prov., Bayingxiang, 2004.Ⅷ.16, coll. Cong Wei and Meixia Yang; 1 ♂, China, Yunnan Prov., Yaoqu, 2009.Ⅵ.03, coll. Wei Cui.

**Distribution.** China (Yunnan, Jiangxi); Thailand.

##### *Eurhadina* (*Singhardina*) *flavicorona* Cai et Kuoh, 1993

*Eurhadina* (*Singhardina*) *flavicorona* Cai et Kuoh, 1993: 224, Figure 3 [9].

**Material examined.** 1 ♂, China, Fujian Prov., Tongmu, 2010.VII.16, coll. Juan Han.

**Distribution.** China (Fujian, Hunan).

##### *Eurhadina* (*Singhardina*) *flaviscutella* Zhang et Huang **sp. nov.**

(Figure 16A–L)

Body and face brownish. Crown creamy with four orange patches centrally. Pronotum brownish with two pairs of round patches symmetrically anteriorly, central patches orange and lateral patches brownish; several small brownish patches centrally. Scutellum yellow with basal triangles orange and apex blackish. Forewing brownish pale, color patterns as shown in Figures 206 and 207.

**Male genitalia:** Upper lobe of pygofer side long without sclerotized protrusion dorsally; disc of lower lobe with several fine microsetae. Subgenital with long macroseta near basal 1/3; five moderate macrosetae near subapex and four rigid microsetae on apex. Paramere straight and hooked caudally with four fine microsetae on caudal part. Aedeagal shaft slim with a small dorsal protrusion terminally and a pair of bifurcate dorsal appendages; upper branches of dorsal appendages longer and arched toward outer side, but terminally pointed toward each other; lower branches of dorsal appendages bent basad and widened apically.

**Measurement.** Length of male 3.62–3.81 mm (including wings).

**Material examined.** Holotype: ♂, China, Fujian Prov., Mt. Wuyi, 2008.VIII.17, coll. Xia Gao and Xiaoting Li. Paratypes: 43 ♂, same data as holotype.

**Etymology.** The specific epithet is derived from the combined Latin word “*flavus*” and “*scutella*” referring to the distinctly yellow scutellum.

**Notes.** This new species resembles *E.* (*S.*) *extensa*
**sp. nov.** in external characteristics, but differs by its yellow scutellum and having just one pair of bifurcate dorsal appendages on aedeagal shaft.

##### *Eurhadina* (*Singhardina*) *fusca* Cai et Kuoh, 1993

*Eurhadina* (*Singhardina*) *fusca* Cai et Kuoh, 1993: 222, Figure 1 [9].

**Distribution.** China (Yunnan).

##### *Eurhadina* (*Singhardina*) *gracilifurca* Zhang et Huang **sp. nov.**

(Figure 17A–M)

Body light yellowish brown. Face brownish black. Crown sordid white with orange fascia posteriorly. Pronotum creamy with two orange patches anteriorly and a brownish patch centrally; basal triangles yellowish; scutellum creamy with apex blackish and a yellowish patch centrally. Forewing brownish orange, without patch on basal half of corium; brochosome area orange with a brownish broad oblique stripe along distal margin; RP vein with a brown spot centrally; lateral margin of apical cell and its adjoining partial region covered with irregular brownish patches.

**Male genitalia:** Upper lobe of pygofer side serrated with a rigid microsetae distally; lower lobe slightly pigmented. Subgenital with long macroseta near basal 1/3; five moderate macrosetae near apical 1/3 and four rigid microsetae on apex. Connective long with lateral arms weakly separated. Paramere with four sensory pores and three fine microsetae on caudal part. Aedeagal shaft slightly expanded centrally with four pairs of apical appendages; upper dorsal appendages slender and Y-shaped, lower dorsal appendages wide basally and V-shaped in dorsal view; both pairs of ventral appendages shorter, slender and merged basally.

**Measurement.** Length of male 2.91–3.34 mm (including wings).

**Material examined.** Holotype: ♂, China, Yunnan Prov., Sanchahe, 1991.VI.7, coll. Rungang Tian. Paratypes: 7 ♂, same data as holotype.

**Etymology.** The specific epithet is derived from the combination of the Latin words “*gracilis*” and “*furca*” that refers to dorsal appendages of aedeagal shaft like a slender long fork and distinctly higher than other appendages.

**Notes.** This new species resembles *E.* (*S.*) *choui* in the color pattern of forewing, but it can be distinguished from the latter by aedeagal shaft with two pairs of ventral appendages.

##### *Eurhadina* (*Singhardina*) *lata* Zhang et Huang **sp. nov.**

(Figure 18A–M)

Body and face reddish orange. Crown with a creamy longitudinal fascia, remainder reddish orange. Pronotum with a creamy longitudinal fascia and two triangular patches posteriorly, remainder reddish orange. Basal triangles pale yellowish, scutellum creamy with apex blackish, two blackish marks on sides of scutellum and an orange patch centrally. Forewing orange with a blackish patch on central part of claval; a longitudinal pink stripe on corium, parallel to the other oblique stripe along claval furrow; brochosome area yellowish orange with a broad oblique brownish stripe along distal margin; apical cell covered with irregular brownish-yellow patches.

**Male genitalia:** Lobe of pygofer side pigmented terminally, upper lobe serrated and lower lobe with a horn-like appendage distally and disc of lower lobe with several rigid microsetae. Subgenital with long macroseta near basal 1/3; two short macrosetae near subapex and three rigid microsetae on apex. Paramere distinctly expanded caudally with three sensory pores and five fine microsetae on caudal part. Aedeagal shaft slim and with two pairs of apical appendages; both dorsal and ventral appendages pointed towards outside in dorsal view; ventral appendage shorter than dorsal ones and slightly expanded subapically with short and sinuated upper branches near basal 1/3.

**Measurement.** Length of male 2.86–3.09 mm (including wings).

**Material examined.** Holotype: ♂, China, Yunnan Prov., Sanchahe, 1991.VI.7, coll. Rungang Tian. Paratypes: 11 ♂, same data as holotype.

**Etymology.** The specific epithet is derived from the Latin word “*latus*”, referring to its paramere expanding apically.

**Notes.** This new species resembles *E.* (*S.*) *dazhulana* in external characteristics and male genitalia, but it can be differentiated by the shorter and sinuated upper branches of ventral appendages of aedeagal shaft in dorsal view.

##### *Eurhadina* (*Singhardina*) *nasti* Dworakowska, 1969

*Eurhadina* (*Singhardina*) *nasti* Dworakowska, 1969: 76, Figures 29–31, 37, 38, 45, 53, 56, 97, 98, 106 [2].

**Distribution.** China (Guangdong).

##### *Eurhadina* (*Singhardina*) *rubrania* Huang et Zhang, 1999

*Eurhadina* (*Singhardina*) *rubrania* Huang et Zhang, 1999: 246, Figure 1 [8].

**Material examined.** 5 ♂, China, Zhejiang Prov., Mt. Gutian, 2003.VIII.17, coll. Wu Dai; 2 ♂, China, Fujian Prov., Mt. Fengyang, 2003.VIII.17, coll. Wu Dai.

**Distribution.** China (Zhejiang, Hunan).

##### *Eurhadina* (*Singhardina*) *rubrivittata* Chiang, Hsu et Knight, 1989

*Eurhadina* (*Singhardina*) *rubrivittata* Chiang, Hsu et Knight, 1989: 125, Figure 14 [13].

**Distribution.** China (Taiwan).

##### *Eurhadina* (*Singhardina*) *rutilans* Hu et Kuoh, 1991

(Figure 19A–M)

*Eurhadina* (*Singhardina*) *rutilans* Hu et Kuoh, 1991: 260, Figure 6 [10].

**Material examined.** 25 ♂, China, Yunnan Prov., Sanchahe, 1991.VI.7, coll. Rungang Tian; 1 ♂, China, Yunnan Prov., Yaoqu, 2009.V.28, coll. Wei Cui; 3 ♂, China, Yunnan Prov., Yaoqu, 2009.Ⅵ.03, coll. Wei Cui; 1 ♂, China, Yunnan Prov., Yaoqu, 2009.Ⅵ.01, coll. Wei Cui.

**Distribution.** China (Yunnan); Thailand.

##### *Eurhadina* (*Singhardina*) *spinifera* Huang et Zhang, 1999

(Figure 20A–L)

*Eurhadina* (*Singhardina*) *spinifera* Huang et Zhang, 1999: 248, Figure 3 [8].

**Material examined.** 3 ♂, China, Fujian Prov., Sangang, 1980.IX.17, coll. Tong Chen; 3 ♂, China, Fujian Prov., Sangang, 1984.VIII.08, coll. Zhixin Cui; 13 ♂, China, Fujian Prov., Mt. Wuyi, 2008.VIII.17, coll. Xia Gao and Xiaoting Li.

**Distribution.** China (Fujian).

##### *Eurhadina* (*Singhardina*) *tripunctata* Huang et Zhang, 1999

*Eurhadina* (*Singhardina*) *tripunctata* Huang et Zhang, 1999: 254, Figure 8 [8].

**Material examined.** 1 ♂, China, Fujian Prov., Huangkeng, 1974.X.25, coll. Fasheng Li; 3 ♂, China, Fujian Prov., Taishun, 2021.VII.10, coll. Juan Zhou.

**Distribution.** China (Fujian)

##### *Eurhadina* (*Singhardina*) *unilobata* Chiang, Hsu et Knight, 1989

*Eurhadina* (*Singhardina*) *unilobata* Chiang, Hsu et Knight, 1989: 125 [13]; Dworakowska, 2002: 94 [3].

**Distribution.** China (Taiwan).

##### *Eurhadina* (*Singhardina*) *uprotrusa* Zhang et Huang **sp. nov.**

(Figure 21A–M)

Body yellowish ochre. Face sordid white. Crown, pronotum, and scutellum yellowish ochre; basal triangles blackish, scutellum with a blackish patch on apex. Forewing pale white with pale pinkish patch on basal part of corium and clavus; brochosome area yellowish with a big brownish patch on costal margin and an oblique brownish stripe along distal margin; apical cell and its adjoining area covered with irregular brownish pale patches.

**Male genitalia:** Caudal margin of upper lobe of pygofer side with several rigid microsetae laterally, lower lobe not pigmented. Subgenital with long macroseta near basal 1/4; three short microsetae extend from apical 1/4 to subapex and with five rigid microsetae on apex. Connective long with lateral arms weakly separated. Paramere blunt terminally with four sensory pores and three fine microsetae on caudal part. Aedeagal shaft slim and curved in lateral view with two pairs of unbranched and slender apical appendages; dorsal appendages shorter than ventral ones, and U-shaped; ventral appendages V-shaped in dorsal view.

**Measurement.** Length of male 3.01–3.25 mm (including wings).

**Material examined.** Holotype: ♂, China, Yunnan Prov., Yaoqu, 2009.VI.03, coll. Wei Cui. 9 ♂, China, Yunnan Prov., Bubang, 2010.IX.04, coll. Meng Zhang.

**Etymology.** The specific epithet is derived from the combination of the letter “*u*” with the Latin word “*protrusus*” that refers to its dorsal appendages of aedeagal shaft U-shaped in dorsal view.

**Notes.** This new species resembles *E.* (*S.*) *uszata* in external characters, but it can be distinguished from the latter by aedeagal shaft with dorsal appendages U-shaped and ventral appendages V-shaped in dorsal view.

##### *Eurhadina* (*Singhardina*) *wuyiana* Yang et Li, 1991

(Figure 22A–N)

*Eurhadina* (*Singhardina*) *wuyiana* Yang et Li, 1991: 26, Figures 19–24 [12].

*Eurhadina flavescens* Huang et Zhang, 1999: 250, Figure 4 [8], **syn. nov.**

**Material examined.** 1 ♂ (holotype), China, Fujian Prov., Chongan, Sangang, 1979.VI.27, coll. Jikun Yang; 2 ♂, China, Fujian Prov., Sangang, 1979.VI.27, coll. Jikun Yang; 1 ♂, China, Sichuan Prov., Moxi, 1999.XI.4, coll. Irena Dworakowska; 1 ♂ China, Yunnan Prov., Tengchong, 1999.XI.22, coll. Irena Dworakowska; 16 ♂, China, Yunnan Prov., Sanchahe, 1991.VI.7, coll. Rungang Tian; 1 ♂, China, Fujian Prov., Mt. Wuyi, 2008.VIII.16, coll. Bin Xiao; 2 ♂, China, Fujian Prov., Mt. Wuyi, 2008.VIII.16, coll. Xia Gao and Xiaoting Li; 30 ♂, China, Fujian Prov., Mt. Wuyi, 2008.VIII.17, coll. Xia Gao and Xiaoting Li; 4 ♂, China, Zhejiang Prov., Wuyanling, 2005.VII.30, coll. Yani Duan; 10 ♂, China, Zhejiang Prov., Wuyanling, 2005.VII.28, coll. Yani Duan; 1 ♂, China, Zhejiang Prov., Wuyanling, 2005.VIII.01, coll. Yani Duan; 1 ♂, China, Zhejiang Prov., Wuyanling, 2005.VIII.02, coll. Yani Duan; 1 ♂, China, Fujian Prov., Mt. Fengyang, 2003.VIII.08, coll. Wu Dai; 1 ♂, China, Zhejiang Prov., Province, 2003.VIII.13, coll. Wu Dai.

**Distribution.** China (Zhejiang, Fujian, Hunan, Sichuan, Yunnan).

**Notes.** Comparing the habitus, detected genitalia, and illustrated figures of the holotype specimen of *E.* (*S.*) *wuyiana* Yang et Li [12] collected in Fujian with *Eurhadina flavescens* as described by Huang et Zhang in 1999 [8] and collected in Fujian and Hunan, both presented identical characteristics: yellowish-orange body, scutellum with two lateral blackish spots and apex blackish, the aedeagus with two pairs of apical appendages; dorsal appendages triforked apically with three branches equal in length; ventral ones slender and simple. So, we find them identical and, according to the rule of ICZN, we propose *Eurhadina flavescens* Huang et Zhang as a junior synonym to *E.* (*S.*) *wuyiana* Yang et Li.

##### *Eurhadina* (*Singhardina*) *zadyma* Dworakowska, 2002, **rec. nov.**

(Figure 23A–L)

*Eurhadina* (*Singhardina*) *zadyma* Dworakowska, 2002: 49, Figures 18–27 [3].

**Material examined.** 7 ♂, China, Yunnan Prov., Longmen, 2009.V.18, coll. Wei Cui.

**Distribution.** China (Yunnan); Thailand.

#### 3.3.4. The *Eurhadina* (*Singhardina*) *mamata* Species Group from China

*E.* (*S.*) *acapitata*, *E.* (*S.*) *flavistriata*, *E.* (*S.*) *jarrayi*
**rec. nov.**, *E.* (*S.*) *prima*
**rec. nov.**, *E.* (*S.*) *quadrimacularis*
**sp. nov.** and *E.* (*S.*) *yingfengica*.

**Description.** Coloration sordid whitish. Crown and pronotum usually with yellow or orange patches or markings anteriorly; pronotum usually with brown markings posteriorly. Scutellum with or without a pair of black preapical spots. Clavus and adjacent part of corium usually with large beige to brown area; RP vein with a brownish spot centrally, costal margin of forewing and apical cells with various beige to brown shades.

Pygofer with large dorsal lobe and small ventral one; both lobes or the ventral ones pigmented distally; upper lobe usually tuberculated and serrated with several rigid microsetae terminally. Subgenital plate with narrowed apex not distinctly extended and with a row of moderate or short macrosetae extending from apical 1/3 to subapex. Paramere with caudal part distinctly shorter than combined length of anterior and central parts, with row of sensory pores on inner margin and fine microsetae on outer margin. Connective short. Penis usually with single dorsal appendages and bifurcate ventral appendages which along with lower branches shorter (lower branch reduced in *E.* (*S.*) *prima*, but longer in *E.* (*S.*) *acapitata*).

##### *Eurhadina* (*Singhardina*) *acapitata* Dworakowska, 1982

*Eurhadina* (*Singhardina*) *acapitata* Dworakowska, 1982: 159, Figures 687–693 [14].

**Distribution.** China (Taiwan).

##### *Eurhadina* (*Singhardina*) *flavistriata* Yang et Li, 1991

*Eurhadina* (*Singhardina*) *flavistriata* Yang et Li, 1991: 27, Figures 32–37 [12].

**Material examined.** 1 ♂, China, Zhejiang Prov., Mt. Gutian, 2003.VIII.17, coll. Wu Dai; 3 ♂, China, Zhejiang Prov., Mt. Tianmu, 2003.VIII, coll. Wu Dai; 6 ♂, China, Fujian Prov., Mt. Wuyi, 2008.VIII.16, coll. Xia Gao and Xiaoting Li; 42 ♂, China, Fujian Prov., Mt. Wuyi, 2008.VIII.17, coll. Xia Gao and Xiaoting Li.

**Distribution.** China (Zhejiang, Fujian).

##### *Eurhadina* (*Singhardina*) *jarrayi* Dworakowska, 2002 **rec. nov.**

(Figure 24A–M)

*Eurhadina* (*Singhardina*) *jarrayi* Dworakowska, 2002: 72, Figures 180–188 [3].

**Material examined.** 1 ♂, China, Yunnan Prov., Sanchahe, 1991.VI.7, coll. Rungang Tian; 1 ♂, China, Yunnan Prov., Yaoqu, 7.VI.2009, coll. Wei Cui.

**Distribution.** China (Yunnan); Thailand.

##### *Eurhadina* (*Singhardina*) *prima* Dworakowska, 2002 **rec. nov.**

(Figure 25A–O)

*Eurhadina* (*Singhardina*) *prima* Dworakowska, 2002: 81, Figures 248–258 [3].

**Material examined.** 1 ♂, China, Yunnan Prov., Sanchahe, 1991.VI.7, coll. Rungang Tian; 1 ♂, China, Yunnan Prov., Yaoqu, 7.VI.2009, coll. Wei Cui.

**Distribution.** China (Yunnan); Thailand.

##### *Eurhadina* (*Singhardina*) *quadrimacularis* Zhang et Huang **sp. nov.**

(Figure 26A–M)

Crown with two reddish orange patches anteriorly. Pronotum with two reddish-orange patches anteriorly and brownish-yellow fascia posteriorly. Scutellum with a pair of black preapical spots. Forewing with a longitudinal yellowish-brown streak on corium and claval along claval furrow; brochosome area whitish with a short oblique brownish-yellow stripe along distal margin; first apical cell and its adjoining area covered with irregular brownish patches.

**Male genitalia:** Both lobes of pygofer side pigmented terminally; lower lobe with a finger-like appendage and several fine microsetae distally. Subgenital plate with long macroseta near basal 1/3, three moderate and four short macrosetae extend from apical 1/3 to subapex and several rigid microsetae on apex. Paramere slender, caudal part with four sensory pores on inner margin and two fine microsetae on outer margin. Connective slender with central ridge long and lateral arms weakly separated. Aedeagal shaft slim and slightly curved in lateral view with two pairs of apical appendages; dorsal appendage slender and shorter than ventral ones; ventral appendages wide and slightly twisted subapically with a short and slender branch basally.

**Measurement.** Length of male 3.35–3.52 mm (including wings).

**Specimens examined.** Holotype, ♂, China, Yunnan Prov., Sanchahe, 1991.VII.7, coll. Rungang Tian. Paratypes, 1 ♂, same data as holotype; 1 ♂, China, Yunnan Prov., Yaoqu, 1999.XII.16, coll. I. Dworakowska; 1 ♂, Chuxiong, Yunnan Province, 1986.VIII.8, coll. Leyi Zheng.

**Etymology.** The specific epithet is derived from the combined Latin prefix “*quadri-*” and “*macula*” referring to its crown and pronotum having four patches.

**Notes.** This new species most resembles *E.* (*S.*) *mamata* in external characters and male genitalia, but differs from the latter in having two patches on the anterior margin of the crown and being wider with twisted upper branches of ventral appendages of aedeagal shaft.

##### *Eurhadina* (*Singhardina*) *yingfengica* Dworakowska, 2002

*Eurhadina* (*Singhardina*) *yingfengica* Dworakowska, 2002: 77, Figures 223–234 [3].

**Material examined.** 12 ♂, China, Yunnan Prov., Sanchahe, 1991.VI.7, coll. Rungang Tian.

**Distribution.** China (Taiwan, Yunnan).

#### 3.3.5. The *Eurhadina* (*Singhardina*) *robusta* Species Group from China

*E.* (*S.*) *dissimilis*, *E.* (*S.*) *flatilis*, *E.* (*S.*) *fumosa*, *E.* (*S.*) *furca*, *E.* (*S.*) *immatura*, *E.* (*S.*) *krispinilla* and *E.* (*S.*) *pookiewica*.

**Description.** Coloration sordid whitish. Crown and anterior margin of pronotum most often unicolored. Pronotum with colored patterns centrally and posteriorly. Scutum and scutellum with various yellow or ochre markings; scutellum with or without pair of black preapical spots. Forewing with various brown to yellowish-brown shades.

Pygofer short; lower lobe obscured. Subgenital plate hooked apically, with a basal or near basal macroseta; two to several macrosetae longitudinally near apical 1/3 and several rigid microsetae on apex. Paramere with caudal part subequal to combined length of two remaining ones. Connective short with central ridge underdeveloped. Aedeagus usually narrow with two pairs of apical appendages; dorsal appendages often simple, arising from the common stalk and ventral appendages single or bifurcate.

##### *Eurhadina* (*Singhardina*) *dissimilis* Zhang et Huang, 2021

*Eurhadina* (*Singhardina*) *dissimilis* Zhang et Huang, 2021: 512, Figures 26–38 [15].

**Material examined.** 1 ♂ (holotype), China, Yunnan Prov., Sanchahe, 1991.VI.7, coll. Rungang Tian; 2 ♂, same data as holotype; 1 ♂, China, Yunnan Prov., Mengyang, Lianghe, 1991.VI.7, coll. Guangchun Liu and Wanzhi Cai; 1 ♂, China, Yunnan Prov., Yaoqu, 2009.VI.7, coll. Wei Cui.

**Distribution.** China (Yunnan).

##### *Eurhadina* (*Singhardina*) *flatilis* Zhang et Huang, 2021

*Eurhadina* (*Singhardina*) *flatilis* Zhang et Huang, 2021: 512, Figures 39–51 [15].

**Material examined.** 1 ♂ (holotype), China, Yunnan Prov., Sanchahe, 1991.VI.7, coll. Rungang Tian; 35 ♂, same data as holotype; 1 ♂, China, Yunnan Prov., Mengyang, Lianghe, 1991.VI.7, coll. Rungang Tian, Yinglun Wang.

**Distribution.** China (Yunnan).

##### *Eurhadina* (*Singhardina*) *fumosa* Zhang et Huang, 2021

*Eurhadina* (*Singhardina*) *fumosa* Zhang et Huang, 2021: 513, Figures 52–64 [15].

**Material examined.** 1 ♂ (holotype), China, Yunnan Prov., Sanchahe, 1991.VI.7, coll. Rungang Tian; 13 ♂, same data as holotype; 1 ♂, China, Yunnan Prov., Mengyang, Lianghe, 1991.VI.7, coll. Guangchun Liu and Wanzhi Cai.

**Distribution.** China (Yunnan).

##### *Eurhadina* (*Singhardina*) *furca* Zhang et Huang, 2021

*Eurhadina* (*Singhardina*) *furca* Zhang et Huang, 2021: 519, Figures 65–77 [15].

**Material examined.** 1 ♂ (holotype), China, Yunnan Prov. Yaoqu, Mengla, 2009.VI.3, coll. Wei Cui; 28 ♂, China, Yunnan Prov. Yaoqu, Mengla, 2010.IX.5 coll. Juan Han.

**Distribution.** China (Yunnan).

##### *Eurhadina* (*Singhardina*) *immatura* Zhang et Huang, 2021

*Eurhadina* (*Singhardina*) *immatura* Zhang et Huang, 2021: 519, Figures 78–90 [15].

**Material examined.** 1 ♂ (holotype), China, Yunnan Prov., Sanchahe, 1991.VI.7, coll. Rungang Tian; 8 ♂, same data as holotype; 1 ♂, China, Yunnan Prov., Mengla, Mt. Nangong, 1999.Ⅻ.17, coll. Irena Dworakowska.

**Distribution.** China (Yunnan).

##### *Eurhadina* (*Singhardina*) *krispinilla* Dworakowska, 2002

*Eurhadina* (*Singhardina*) *krispinilla* Dworakowska, 2002: 87, Figures 303–310 [3]; Zhou et al., 2021: 512, Figures 1–13 [15].

**Material examined.** 1 ♂ (holotype), China, Yunnan Prov., Sanchahe, 1991.VI.7, coll. Rungang Tian; 26 ♂, same data as holotype; 1 ♂, China, Yunnan Prov., Mengla, 1200 m, 17.XII.1991, coll. Irena Dworakowska; 1 ♂, China, Yunnan Prov., 800 m, 2.VI.2009, coll. Wei Cui.

**Distribution.** China (Yunnan).

##### *Eurhadina* (*Singhardina*) *pookiewica* Dworakowska, 2002

*Eurhadina* (*Singhardina*) *pookiewica* Dworakowska, 2002: 91, Figures 335–344 [3]; Zhou et al., 2021: 512, Figures 14–25 [15].

**Material examined.** 1 ♂ (holotype), China, Yunnan Prov., Sanchahe, 1991.VI.7, coll. Rungang Tian; 28 ♂, same data as holotype.

**Distribution.** China (Yunnan).

### 3.4. Species Diversity in China

So far, the biogeography knowledge of the subgenus is almost unknown. Here, the species geographical distribution and diversity of *Singhardina* in China were presented in Figure 27 based on comprehensive collection data summarized from original descriptions or subsequent literatures.

Yunnan has the richest species diversity with 33 species accounting for 66% of *Singhardina*, In this region, adults can be found from May to December across a narrow range of elevation (800–1200 m), but particularly in summer from May to July. Zhejiang, Fujian, and Hunan have relatively rich species diversity with 7 to 19 species. Only a single species is known from Guangxi, Jiangxi, Guizhou, and Sichuna. *E.* (*S.*) *wuyiana* is the species widest distributed in Zhejiang, Fujian, Hunan, Sichuan, and Yunnan (Figure 27).

## 4. Discussion

Fifty-two species of the leafhopper subgenus *Eurhadina* (*Singhardina*) Mahmood from China are reviewed and divided into five species groups (*E.* (*S.*) *mamata*, *E.* (*S.*) *punjabensis*, *E.* (*S.*) *robusta*, *E.* (*S.*) *rubra*, and *E.* (*S.*) *vittata* groups) based on comparative morphological characteristics, especially by the external characters and male genitalia. Fourteen species new to science are described. Four additional species are newly recorded from China. Two synonymies are proposed: *Eurhadina* (*Singhardina*) *flavescens* Huang et Zhang, 1999 is a junior synonym to *Eurhadina wuyiana* Yang et Li, 1991 and *Eurhadina rubromia* Cai et Kuoh, 1993 is a junior synonym to *Eurhadina* (*Singhardina*) *biavis* Yang et Li, 1991. Altogether, this results in 50 species currently recognized in China.

## 5. Conclusions

According to the geography of both formerly and newly collected species in our museum, we believe that southwest China is a biodiversity hotspot for this genus (as shown in Figure 27). This region should have more new taxa yet to be discovered and may present more future challenges in establishing subgenera and defining species groups.

## Figures and Tables

**Figure 1 insects-13-00345-f001:**
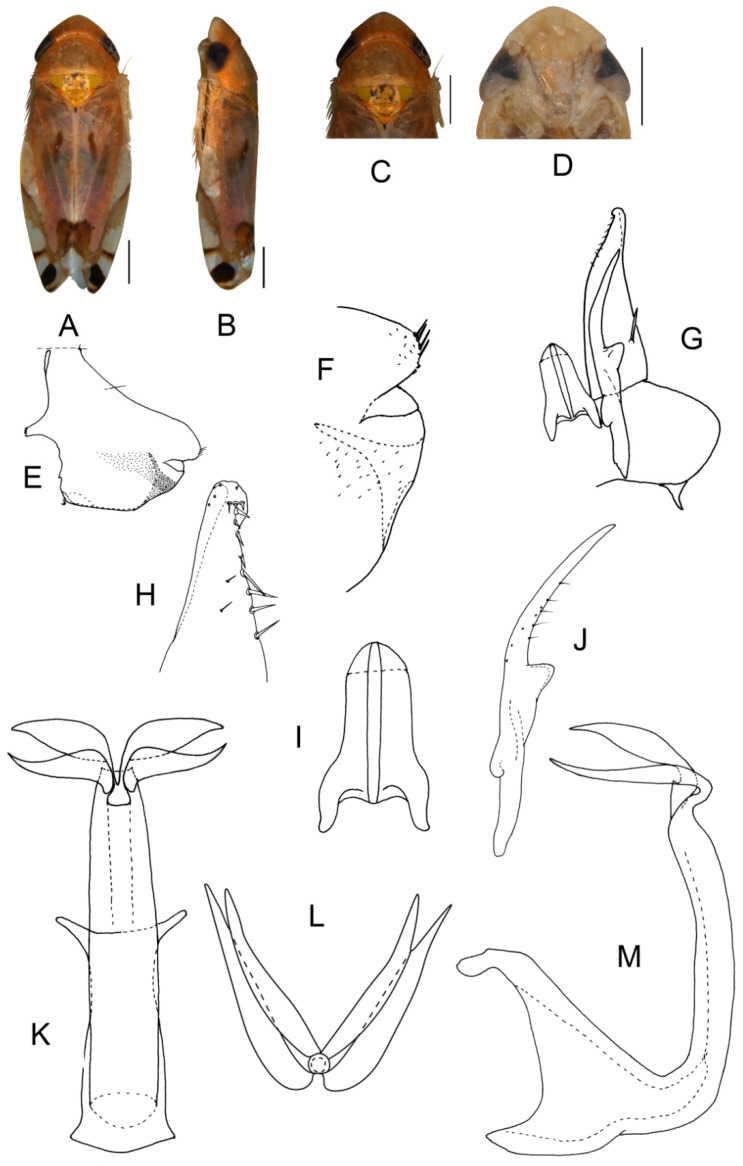
*Eurhadina* (*Singhardina*) *foliiformis*
**sp. nov.**: (**A**) habitus, dorsal view; (**B**) habitus, lateral view; (**C**) head and thorax, dorsal view; (**D**) face; (**E**) male pygofer side, lateral view; (**F**) hind part of pygofer, lateral view; (**G**) subgenital plate, connective, paramere and sternite IX, dorsal view; (**H**) apex of subgenital plate, dorsal view; (**I**) connective, dorsal view; (**J**) paramere, dorsal view; (**K**) aedeagus, posterior view; (**L**) apex of aedeagal shaft, dorsal view; and (**M**) aedeagus, lateral view. Scale bars: 0.5 mm.

**Figure 2 insects-13-00345-f002:**
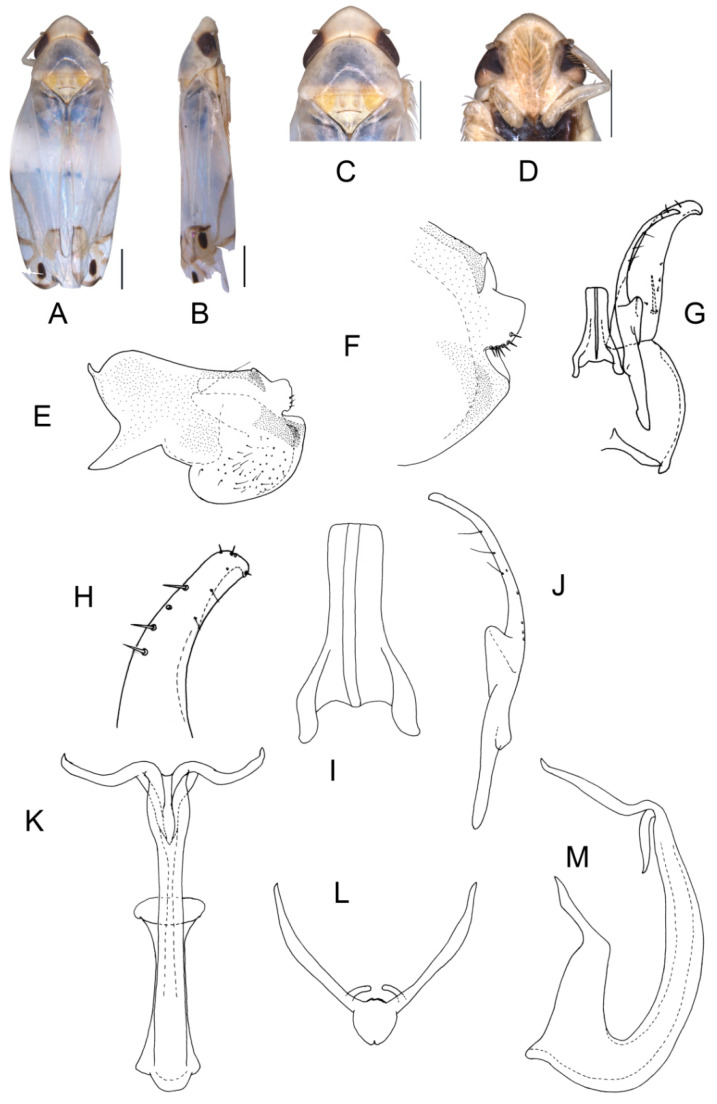
*Eurhadina* (*Singhardina*) *galacta*
**sp. nov.**: (**A**) habitus, dorsal view; (**B**) habitus, lateral view; (**C**) head and thorax, dorsal view; (**D**) face; (**E**) male pygofer side, lateral view; (**F**) hind part of pygofer, lateral view; (**G**) subgenital plate, connective, paramere and sternite IX, dorsal view; (**H**) apex of subgenital plate, dorsal view; (**I**) connective, dorsal view; (**J**) paramere, dorsal view; (**K**) aedeagus, posterior view; (**L**) apex of aedeagal shaft, dorsal view; and (**M**) aedeagus, lateral view. Scale bars: 0.5 mm.

**Figure 3 insects-13-00345-f003:**
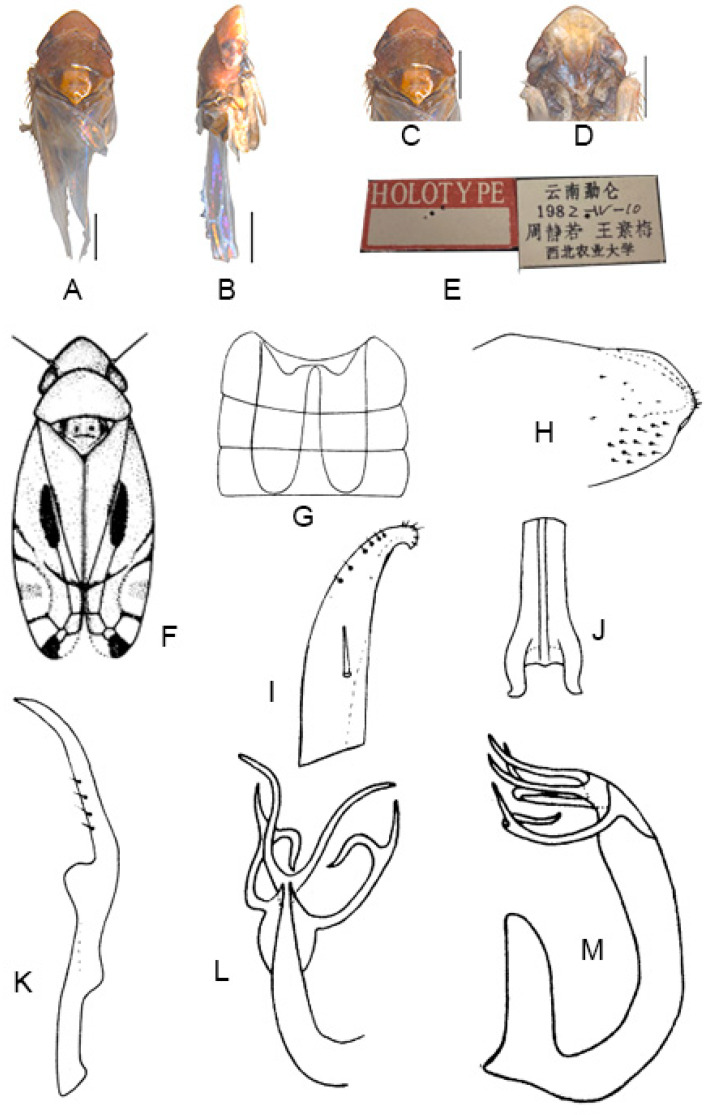
*Eurhadina* (*Singhardina*) *menglunensis*. (**A**–**E**: holotype; **F**–**M**: Huang et Zhang, 1999 [8]) (**A**) habitus, dorsal view; (**B**) habitus, lateral view; (**C**) head and thorax, dorsal view; (**D**) face; (**E**) labels of the holotype (China, Yunnan Prov., Menglun, 1982.IV.10, coll. Jingruo Zhou and Sumei Wang); (**F**) habitus, dorsal view; (**G**) abdominal apodemes, dorsal view; (**H**) male pygofer side, lateral view; (**I**) subgenital plate, dorsal view; (**J**) connective, dorsal view; (**K**) paramere, dorsal view; (**L**) apex of aedeagal shaft, dorsal view; and (**M**) aedeagus, lateral view. Scale bars: 0.5 mm.

**Figure 4 insects-13-00345-f004:**
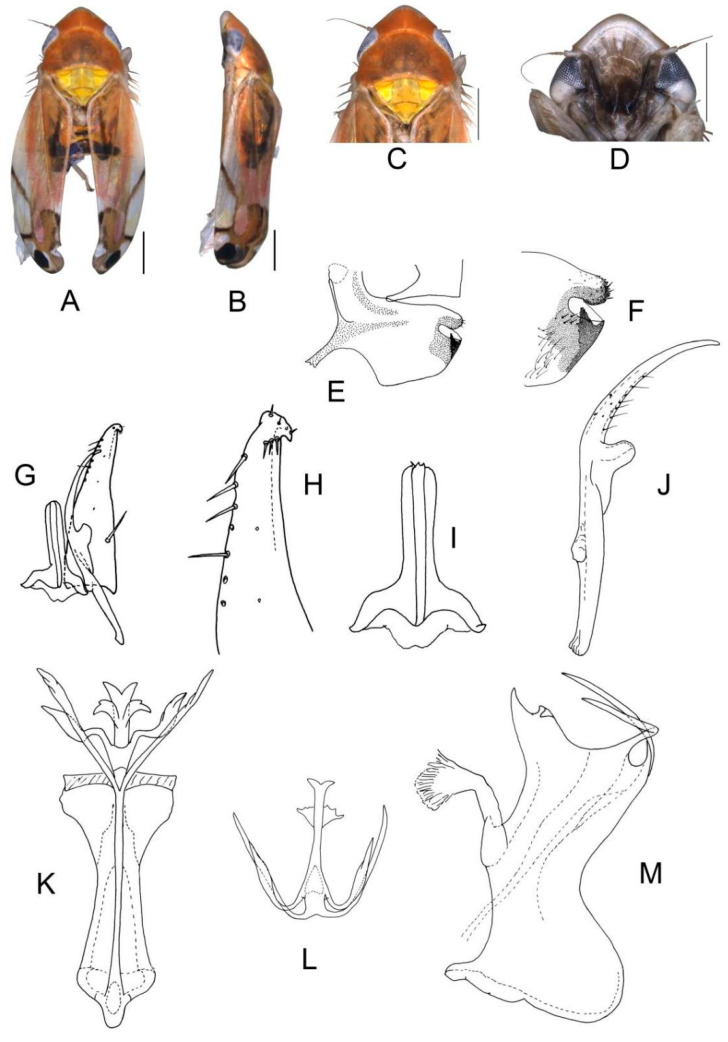
*Eurhadina* (*Singhardina*) *recta*
**sp. nov.**: (**A**) habitus, dorsal view; (**B**) habitus, lateral view; (**C**) head and thorax, dorsal view; (**D**) face; (**E**) male pygofer side, lateral view; (**F**) hind part of pygofer, lateral view; (**G**) subgenital plate, connective and paramere, dorsal view; (**H**) apex of subgenital plate, dorsal view; (**I**) connective, dorsal view; (**J**) paramere, dorsal view; (**K**) aedeagus, posterior view; (**L**) apex of aedeagal shaft, dorsal view; and (**M**) aedeagus, lateral view. Scale bars: 0.5 mm.

**Figure 5 insects-13-00345-f005:**
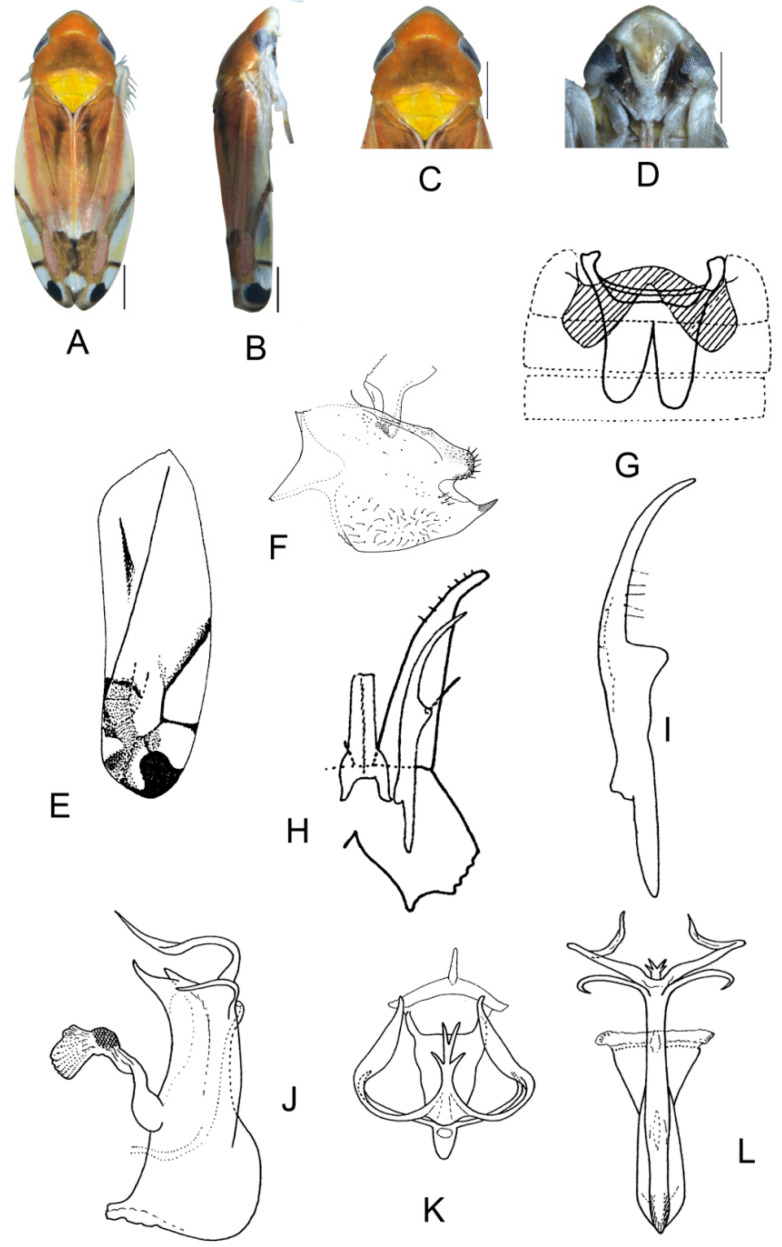
*Eurhadina* (*Singhardina*) *rubra* (**E**–**L**: Dworakowska, 2002 [3]): (**A**) habitus, dorsal view; (**B**) habitus, lateral view; (**C**) head and thorax, dorsal view; (**D**) face; (**E**) forewing; (**F**) male pygofer side, lateral view; (**G**) abdominal apodemes, dorsal view; (**H**) subgenital plate, connective, paramere and sternite IX, dorsal view; (**I**) paramere, dorsal view; (**J**) apex of aedeagal shaft, dorsal view; (**K**) aedeagus, lateral view; and (**L**) aedeagus, posterior view. Scale bars: 0.5 mm.

**Figure 6 insects-13-00345-f006:**
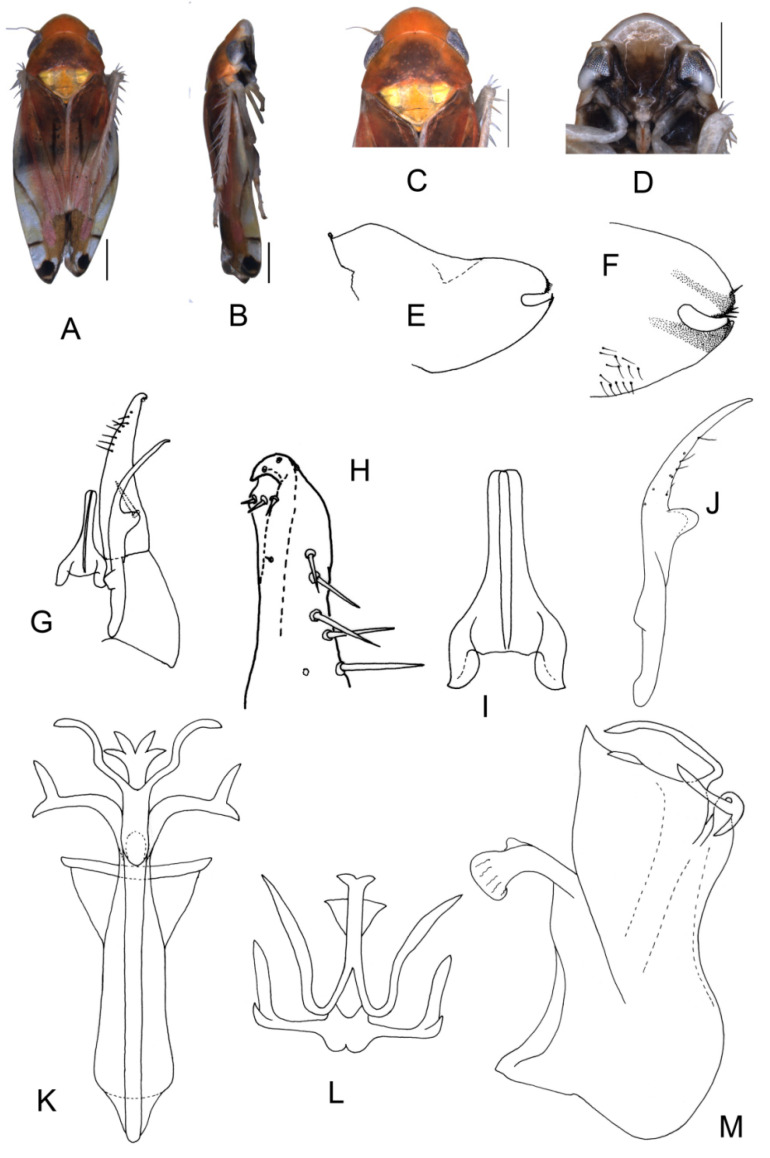
*Eurhadina* (*Singhardina*) *scalesa*
**sp. nov.**: (**A**) habitus, dorsal view; (**B**) habitus, lateral view; (**C**) head and thorax, dorsal view; (**D**) face; (**E**) male pygofer side, lateral view; (**F**) hind part of pygofer, lateral view; (**G**) subgenital plate, connective, paramere and sternite IX, dorsal view; (**H**) apex of subgenital plate, dorsal view; (**I**) connective, dorsal view; (**J**) paramere, dorsal view; (**K**) aedeagus, posterior view; (**L**) apex of aedeagal shaft, dorsal view; and (**M**) aedeagus, lateral view. Scale bars: 0.5 mm.

**Figure 7 insects-13-00345-f007:**
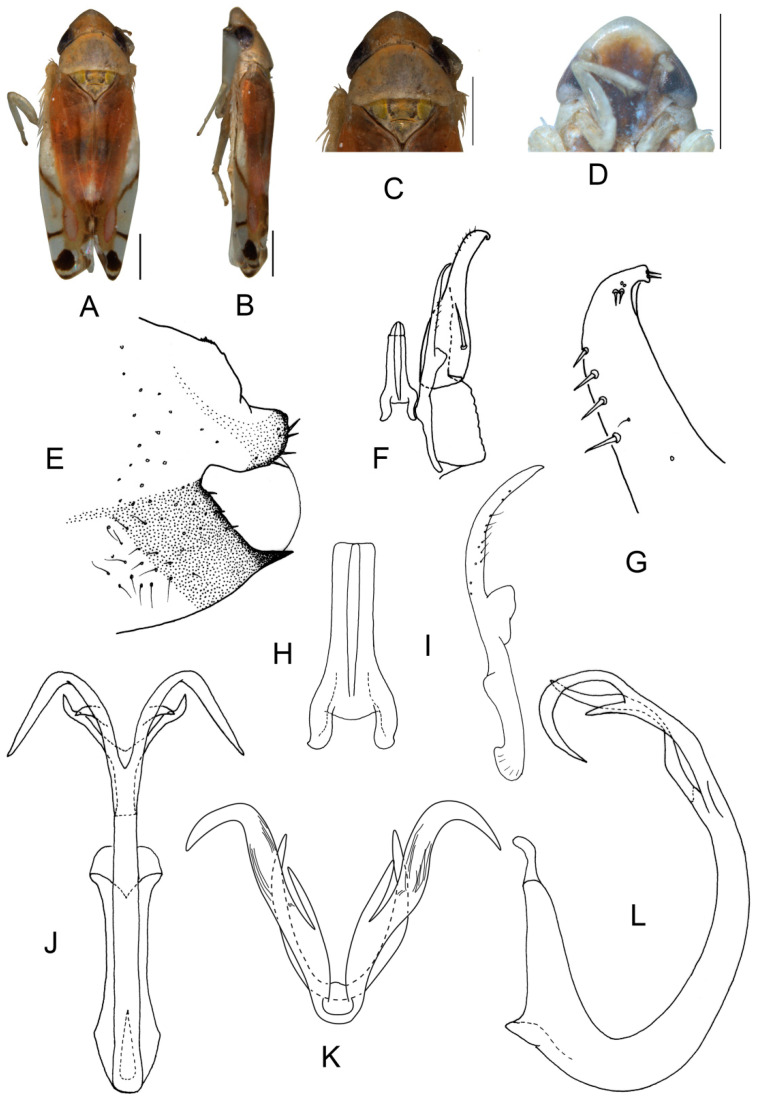
*Eurhadina* (*Singhardina*) *scamba*
**sp. nov.**: (**A**) habitus, dorsal view; (**B**) habitus, lateral view; (**C**) head and thorax, dorsal view; (**D**) face; (**E**) hind part of pygofer, lateral view; (**F**) subgenital plate, connective, paramere and sternite IX, dorsal view; (**G**) apex of subgenital plate, dorsal view; (**H**) connective, dorsal view; (**I**) paramere, dorsal view; (**J**) aedeagus, posterior view; (**K**) apex of aedeagal shaft, dorsal view; and (**L**) aedeagus, lateral view. Scale bars: 0.5 mm.

**Figure 8 insects-13-00345-f008:**
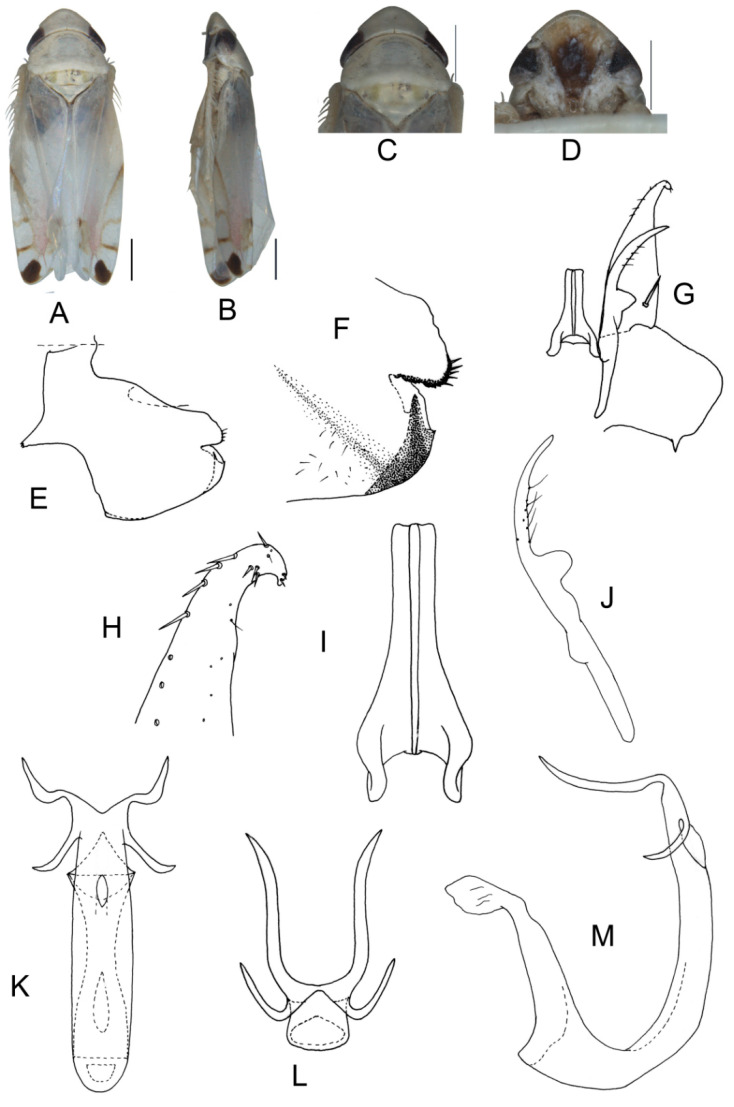
*Eurhadina* (*Singhardina*) *scandens*
**sp. nov.**: (**A**) habitus, dorsal view; (**B**) habitus, lateral view; (**C**) head and thorax, dorsal view; (**D**) face; (**E**) male pygofer side, lateral view; (**F**) hind part of pygofer, lateral view; (**G**) subgenital plate, connective, paramere and sternite IX, dorsal view; (**H**) apex of subgenital plate, dorsal view; (**I**) connective, dorsal view; (**J**) paramere, dorsal view; (**K**) aedeagus, posterior view; (**L**) apex of aedeagal shaft, dorsal view; and (**M**) aedeagus, lateral view. Scale bars: 0.5 mm.

**Figure 9 insects-13-00345-f009:**
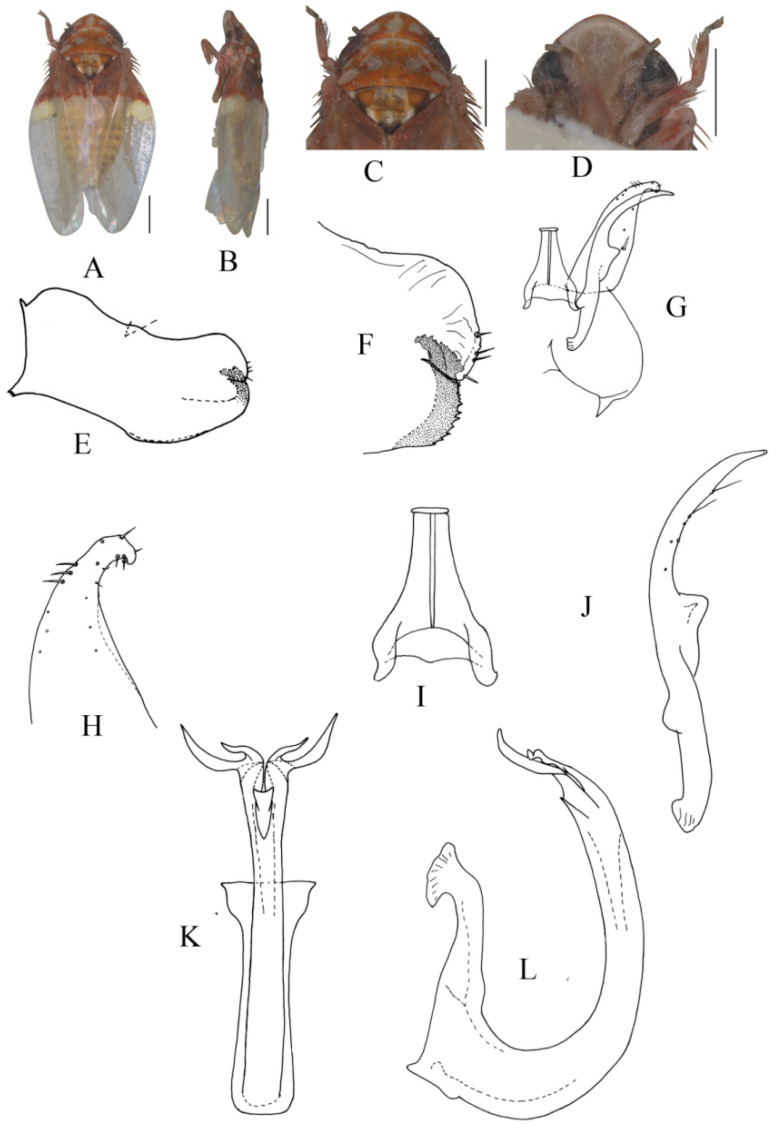
*Eurhadina* (*Singhardina*) *amacularis*
**sp. nov.**: (**A**) habitus, dorsal view; (**B**) habitus, lateral view; (**C**) head and thorax, dorsal view; (**D**) face; (**E**) male pygofer side, lateral view; (**F**) hind part of pygofer, lateral view; (**G**) subgenital plate, connective, paramere and sternite IX, dorsal view; (**H**) apex of subgenital plate, dorsal view; (**I**) connective, dorsal view; (**J**) paramere, dorsal view; (**K**) aedeagus, posterior view; and (**L**) aedeagus, lateral view. Scale bars: 0.5 mm.

**Figure 10 insects-13-00345-f010:**
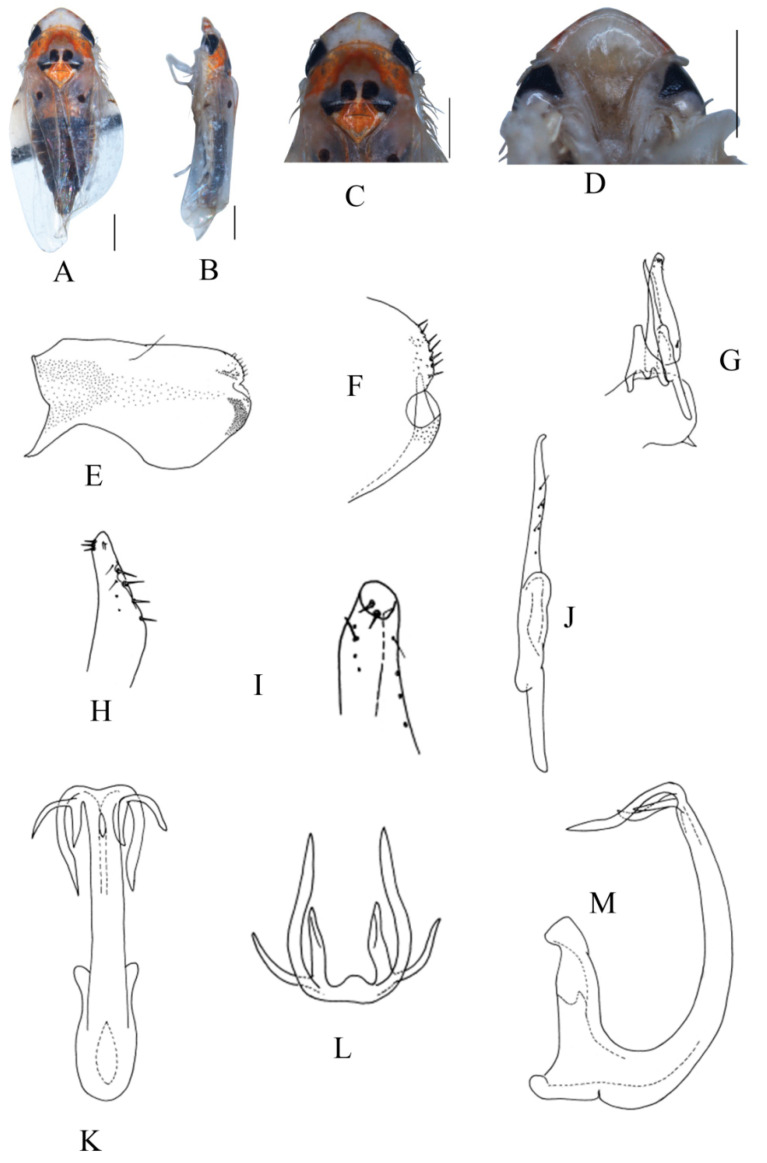
*Eurhadina* (*Singhardina*) *parilintanonica*
**sp. nov.**: (**A**) habitus, dorsal view; (**B**) habitus, lateral view; (**C**) head and thorax, dorsal view; (**D**) face; (**E**) male pygofer side, lateral view; (**F**) hind part of pygofer, lateral view; (**G**) subgenital plate, connective, paramere and sternite IX, dorsal view; (**H**,**I**) apex of subgenital plate, dorsal view; (**J**) paramere, dorsal view; (**K**) aedeagus, posterior view; (**L**) apex of aedeagal shaft, dorsal view; and (**M**) aedeagus, lateral view. Scale bars: 0.5 mm.

**Figure 11 insects-13-00345-f011:**
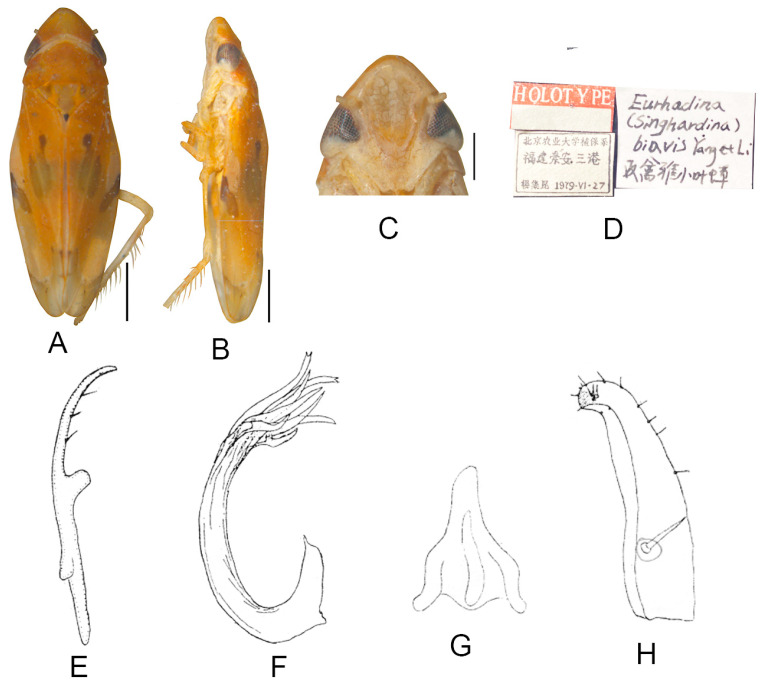
*Eurhadina* (*Singhardina*) *biavis* (**A**–**D**: holotype; **E**–**H**: Yang et Li, 1991 [12]): (**A**) habitus, dorsal view; (**B**) habitus, lateral view; (**C**) face; (**D**) labels of the holotype (China, Fujian Prov., Chongan, Sangang, 1979.VI.27, coll. Jikun Yang); (**E**) paramere, dorsal view; (**F**) aedeagus, lateral view; (**G**) connective, dorsal view; and (**H**) subgenital plate, dorsal view. Scale bars: 0.5 mm.

**Figure 12 insects-13-00345-f012:**
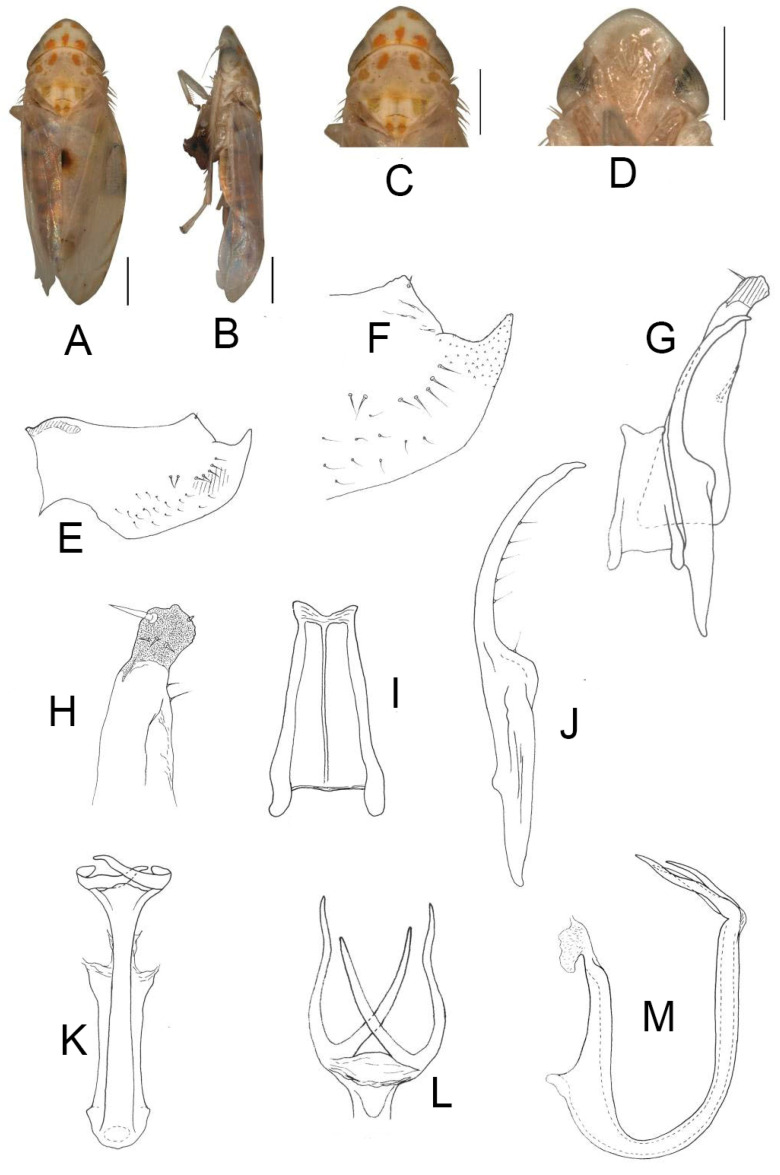
*Eurhadina* (*Singhardina*) *centralis:* (**A**) habitus, dorsal view; (**B**) habitus, lateral view; (**C**) head and thorax, dorsal view; (**D**) face; (**E**) male pygofer side, lateral view; (**F**) hind part of pygofer, lateral view; (**G**) subgenital plate, connective and paramere, dorsal view; (**H**) apex of subgenital plate, dorsal view; (**I**) connective, dorsal view; (**J**) paramere, dorsal view; (**K**) aedeagus, posterior view; (**L**) apex of aedeagal shaft, dorsal view; and (**M**) aedeagus, lateral view. Scale bars: 0.5 mm.

**Figure 13 insects-13-00345-f013:**
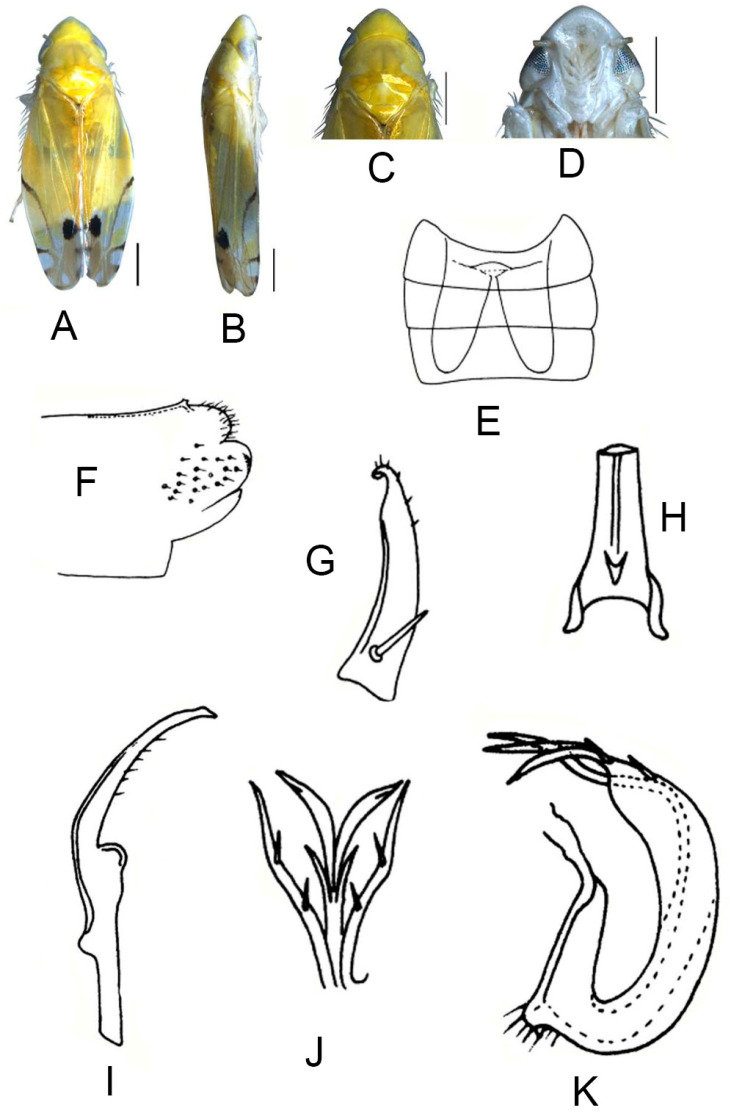
*Eurhadina* (*Singhardina*) *diplopunctata* (**E**–**K**: Huang et Zhang, 1999 [8]): (**A**) habitus, dorsal view; (**B**) habitus, lateral view; (**C**) head and thorax, dorsal view; (**D**) face; (**E**) abdominal apodemes, dorsal view; (**F**) male pygofer side, lateral view; (**G**) subgenital plate, dorsal view; (**H**) connective, dorsal view; (**I**) paramere, dorsal view; (**J**) apex of aedeagal shaft, dorsal view; and (**K**) aedeagus, lateral view. Scale bars: 0.5 mm.

**Figure 14 insects-13-00345-f014:**
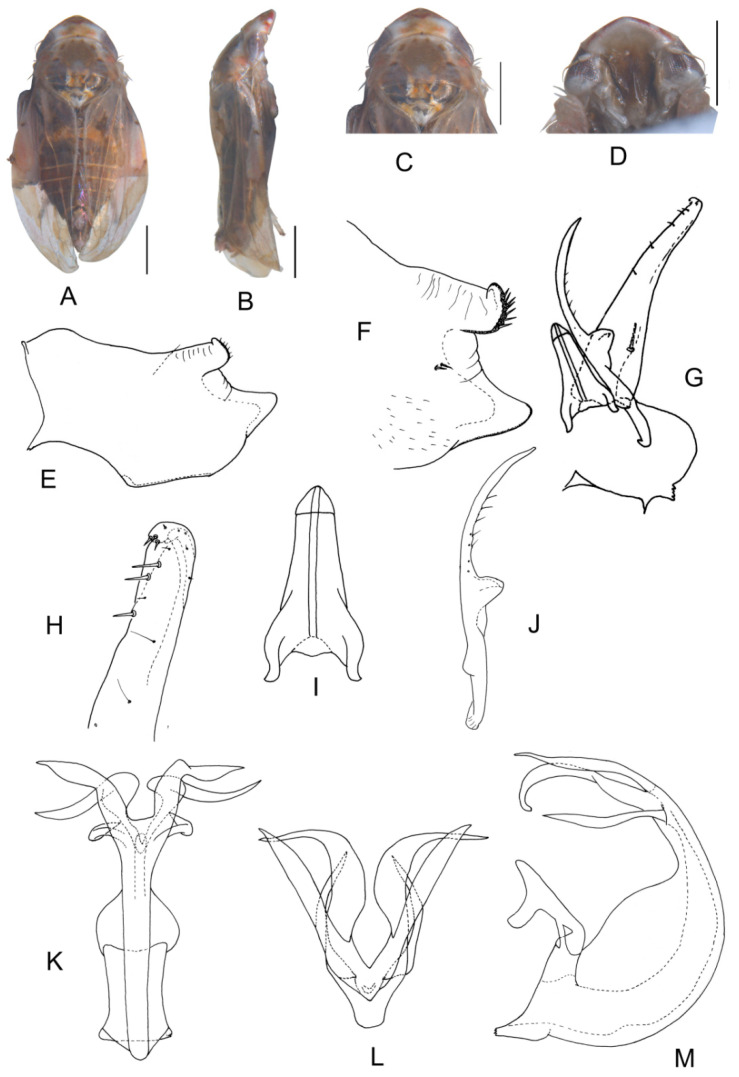
*Eurhadina* (*Singhardina*) *extensa*
**sp. nov.**: (**A**) habitus, dorsal view; (**B**) habitus, lateral view; (**C**) head and thorax, dorsal view; (**D**) face; (**E**) male pygofer side, lateral view; (**F**) hind part of pygofer, lateral view; (**G**) subgenital plate, connective, paramere and sternite IX, dorsal view; (**H**) apex of subgenital plate, dorsal view; (**I**) connective, dorsal view; (**J**) paramere, dorsal view; (**K**) aedeagus, posterior view; (**L**) apex of aedeagal shaft, dorsal view; and (**M**) aedeagus, lateral view. Scale bars: 0.5 mm.

**Figure 15 insects-13-00345-f015:**
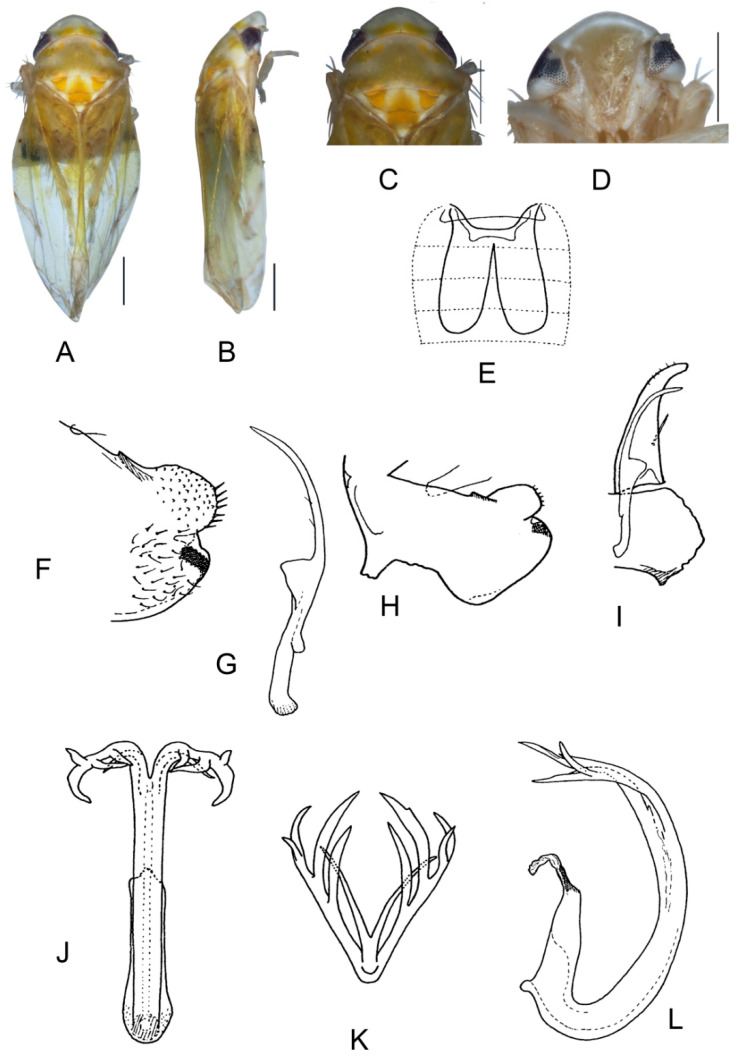
*Eurhadina* (*Singhardina*) *fasciata*
**rec. nov.** (**E**–**L**: Dworakowska, 2002 [3]): (**A**) habitus, dorsal view; (**B**) habitus, lateral view; (**C**) head and thorax, dorsal view; (**D**) face; (**E**) abdominal apodemes, dorsal view; (**F**) hind part of pygofer, lateral view; (**G**) paramere, dorsal view; (**H**) male pygofer side, lateral view; (**I**) subgenital plate, paramere and sternite IX, dorsal view; (**J**) aedeagus, posterior view; (**K**) apex of aedeagal shaft, dorsal view; and (**L**) aedeagus, lateral view. Scale bars: 0.5 mm.

**Figure 16 insects-13-00345-f016:**
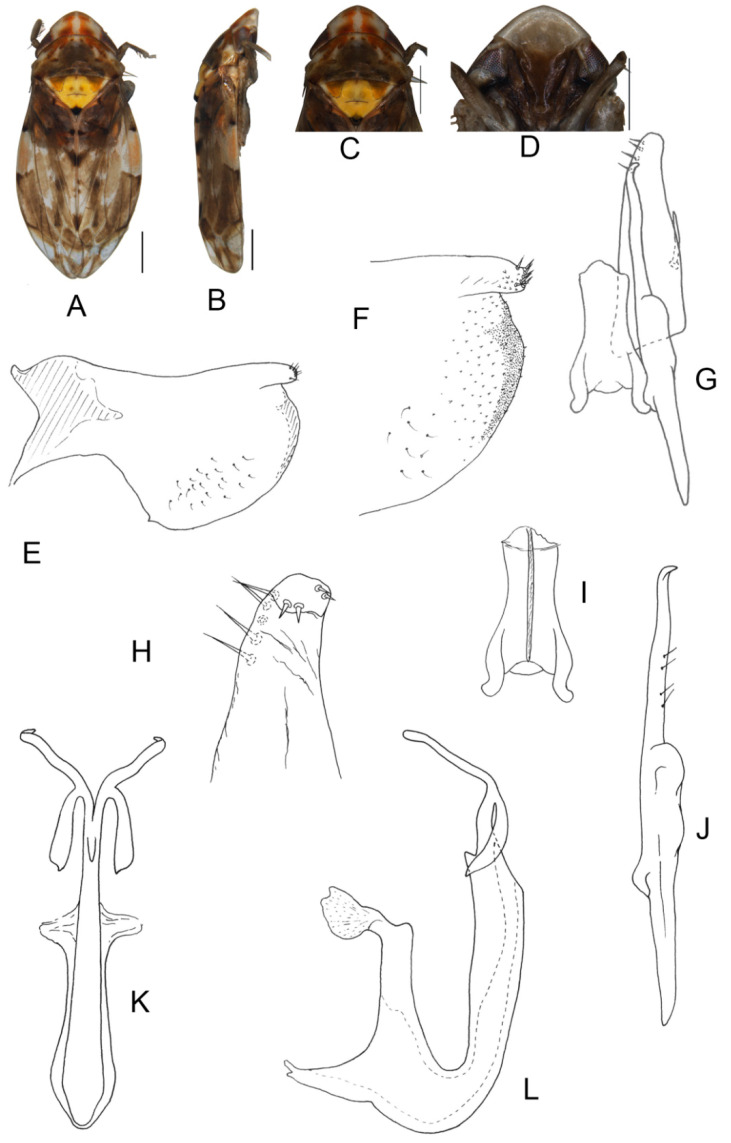
*Eurhadina* (*Singhardina*) *flaviscutella*
**sp. nov.**: (**A**) habitus, dorsal view; (**B**) habitus, lateral view; (**C**) head and thorax, dorsal view; (**D**) face; (**E**) male pygofer side, lateral view; (**F**) hind part of pygofer, lateral view; (**G**) subgenital plate, connective and paramere, dorsal view; (**H**) apex of subgenital plate, dorsal view; (**I**) connective, dorsal view; (**J**) paramere, dorsal view; (**K**) aedeagus, posterior view; and (**L**) aedeagus, lateral view. Scale bars: 0.5 mm.

**Figure 17 insects-13-00345-f017:**
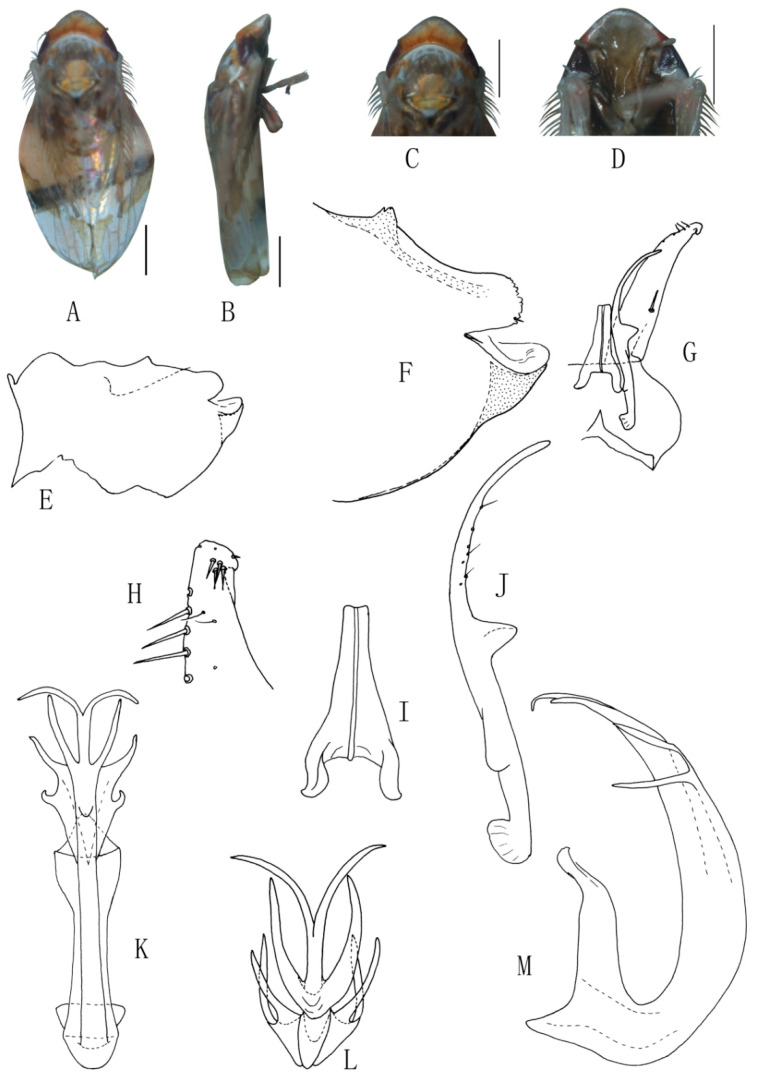
*Eurhadina* (*Singhardina*) *gracilifurca*
**sp. nov.**: (**A**) habitus, dorsal view; (**B**) habitus, lateral view; (**C**) head and thorax, dorsal view; (**D**) face; (**E**) male pygofer side, lateral view; (**F**) hind part of pygofer, lateral view; (**G**) subgenital plate, connective, paramere and sternite IX, dorsal view; (**H**) apex of subgenital plate, dorsal view; (**I**) connective, dorsal view; (**J**) paramere, dorsal view; (**K**) aedeagus, posterior view; (**L**) apex of aedeagal shaft, dorsal view; and (**M**) aedeagus, lateral view. Scale bars: 0.5 mm.

**Figure 18 insects-13-00345-f018:**
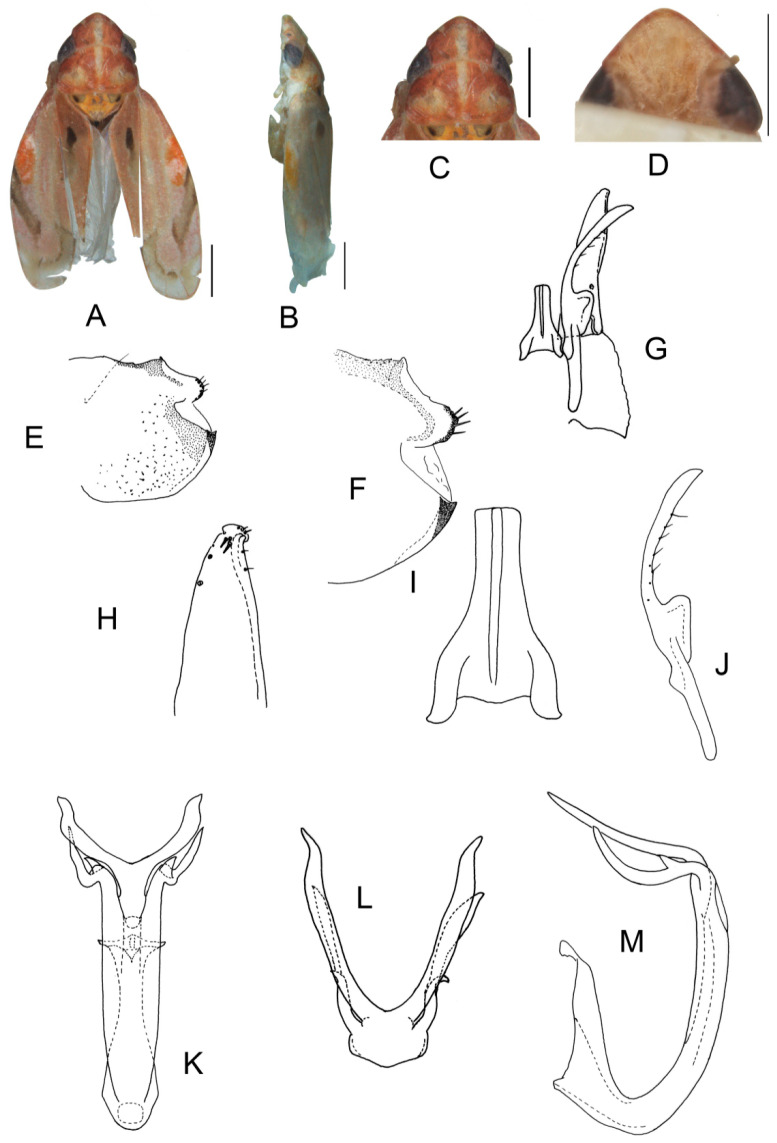
*Eurhadina* (*Singhardina*) *lata*
**sp. nov.**: (**A**) habitus, dorsal view; (**B**) habitus, lateral view; (**C**) head and thorax, dorsal view; (**D**) face; (**E**) male pygofer side, lateral view; (**F**) hind part of pygofer, lateral view; (**G**) subgenital plate, connective, paramere and sternite IX, dorsal view; (**H**) apex of subgenital plate, dorsal view; (**I**) connective, dorsal view; (**J**) paramere, dorsal view; (**K**) aedeagus, posterior view; (**L**) apex of aedeagal shaft, dorsal view; and (**M**) aedeagus, lateral view. Scale bars: 0.5 mm.

**Figure 19 insects-13-00345-f019:**
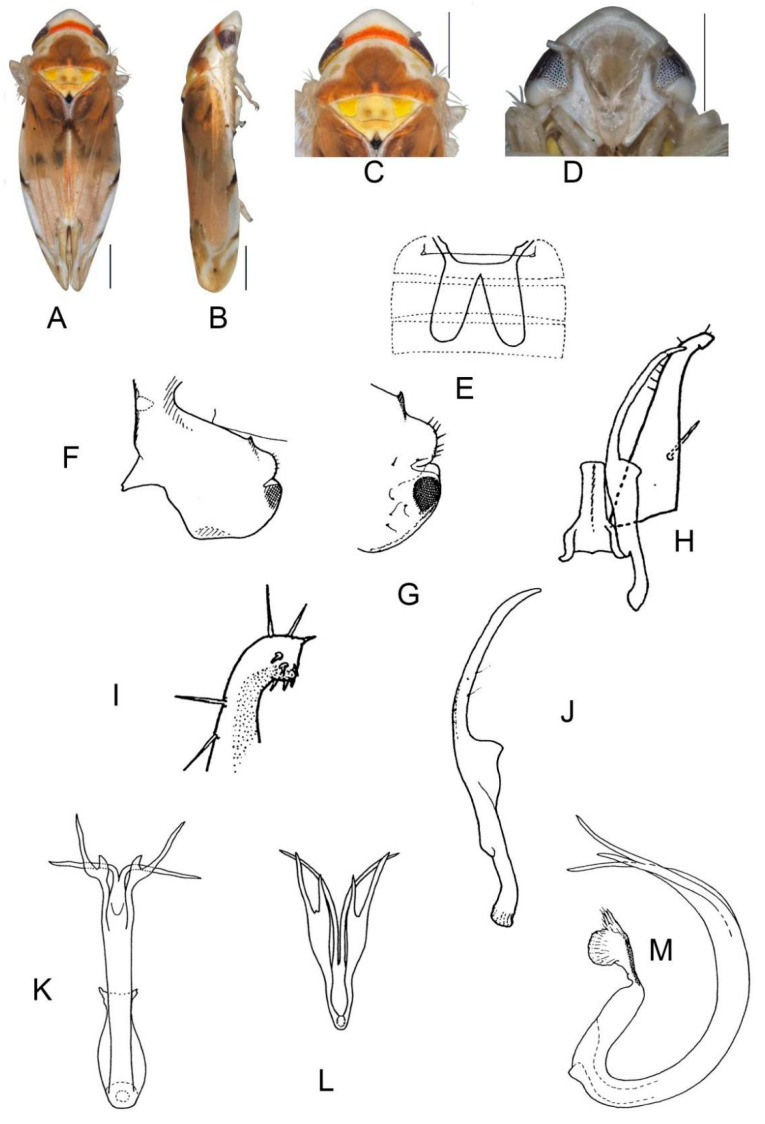
*Eurhadina* (*Singhardina*) *rutilans*: (**A**) habitus, dorsal view; (**B**) habitus, lateral view; (**C**) head and thorax, dorsal view; (**D**) face; (**E**) abdominal apodemes, dorsal view; (**F**) male pygofer side, lateral view; (**G**) hind part of pygofer, lateral view; (**H**) subgenital plate, connective and paramere, dorsal view; (**I**) apex of subgenital plate, dorsal view; (**J**) paramere, dorsal view; (**K**) aedeagus, posterior view; (**L**) apex of aedeagal shaft, dorsal view; and (**M**) aedeagus, lateral view. Scale bars: 0.5 mm.

**Figure 20 insects-13-00345-f020:**
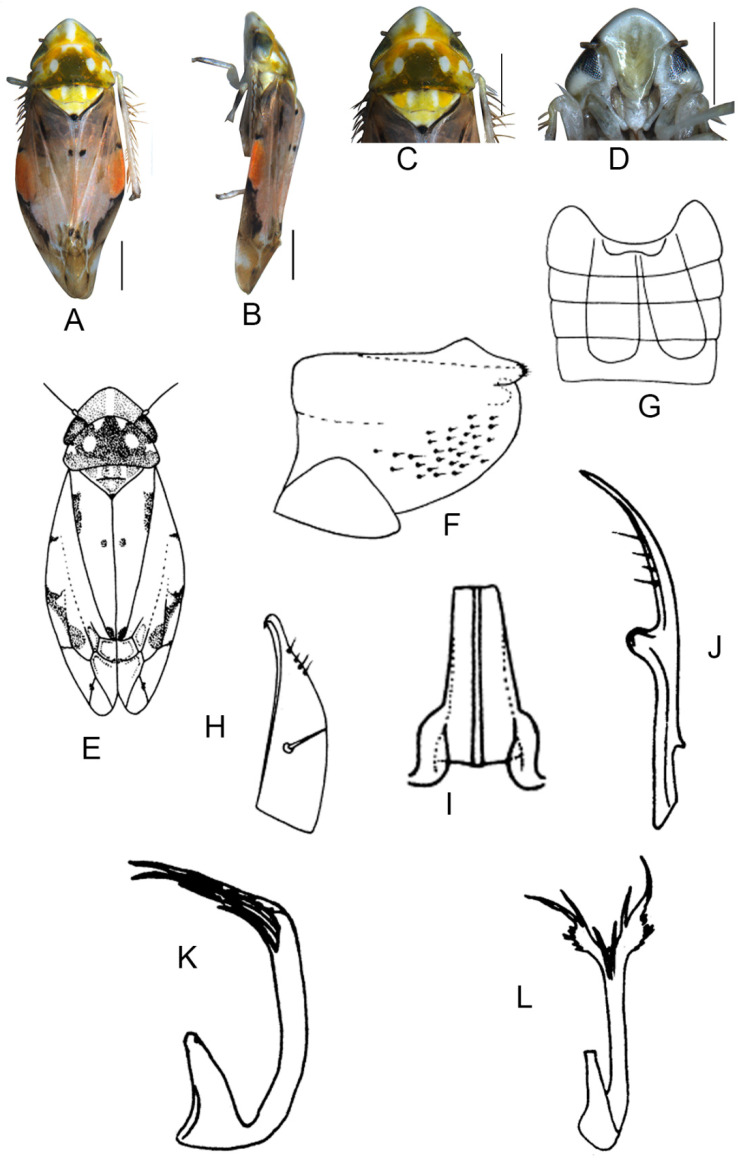
*Eurhadina* (*Singhardina*) *spinifera* (**E**–**L**: Huang et Zhang, 1999 [8]): (**A**) habitus, dorsal view; (**B**) habitus, lateral view; (**C**) head and thorax, dorsal view; (**D**) face; (**E**) habitus, dorsal view; (**F**) male pygofer side, lateral view; (**G**) abdominal apodemes, dorsal view; (**H**) subgenital plate, dorsal view; (**I**) connective, dorsal view; (**J**) paramere, dorsal view; (**K**) aedeagus, lateral view; and (**L**) aedeagus, posterior view. Scale bars: 0.5 mm.

**Figure 21 insects-13-00345-f021:**
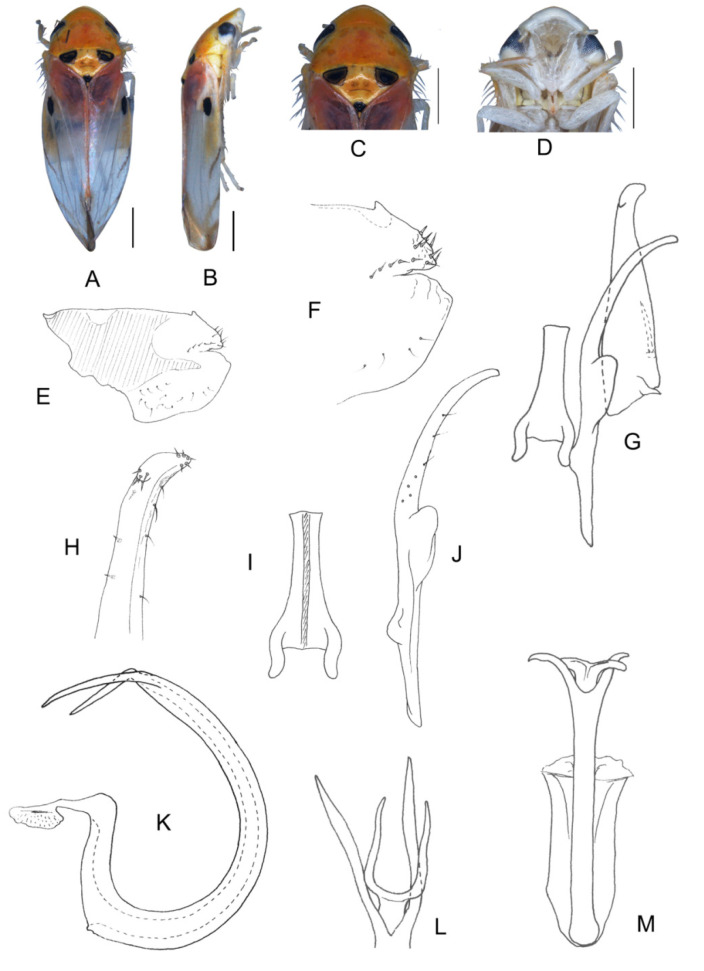
*Eurhadina* (*Singhardina*) *uprotrusa*
**sp. nov.**: (**A**) habitus, dorsal view; (**B**) habitus, lateral view; (**C**) head and thorax, dorsal view; (**D**) face; (**E**) male pygofer side, lateral view; (**F**) hind part of pygofer, lateral view; (**G**) subgenital plate, connective and paramere, lateral view; (**H**) apex of subgenital plate, lateral view; (**I**) connective, lateral view; (**J**) paramere, lateral view; (**K**) aedeagus, posterior view; (**L**) apex of aedeagal shaft, dorsal view; and (**M**) aedeagus, lateral view. Scale bars: 0.5 mm.

**Figure 22 insects-13-00345-f022:**
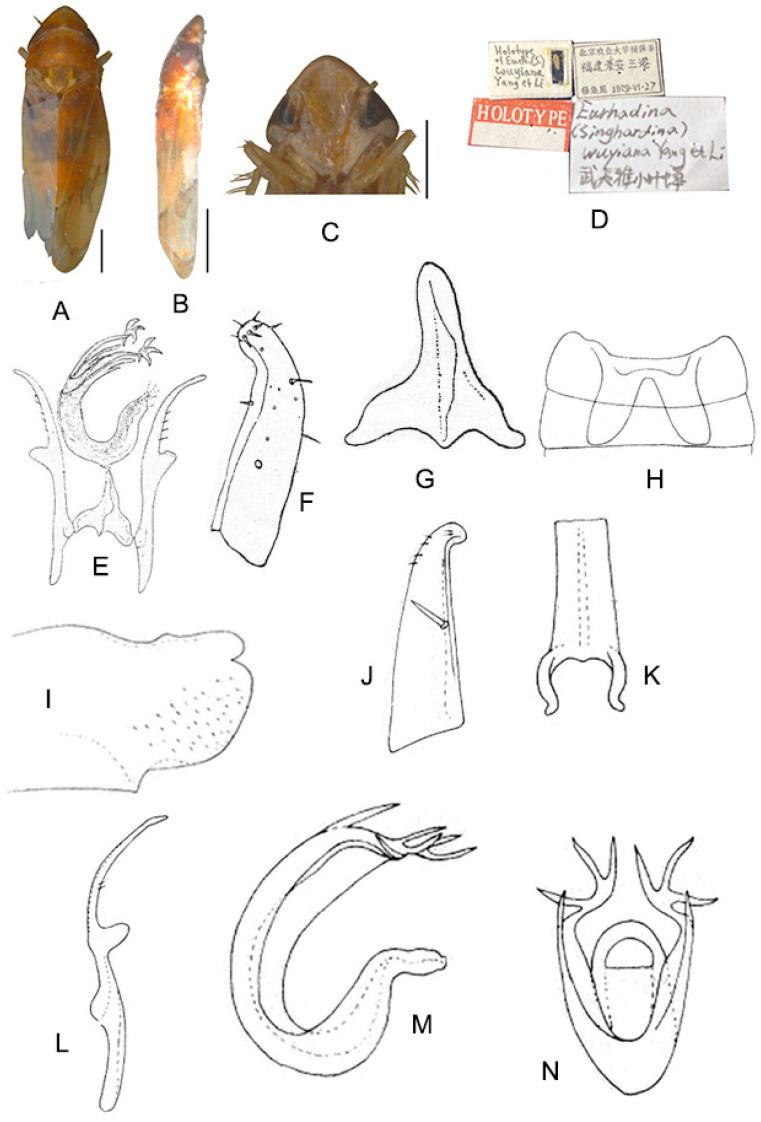
*Eurhadina* (*Singhardina*) *wuyiana* (**A**–**D**: holotype; **E**–**G**: Yang et Li, 1991 [12]; H–N: Huang et Zhang, 1999 [8]); (**A**) habitus, dorsal view; (**B**) habitus, lateral view; (**C**) face; (**D**) labels of the holotype (China, Fujian Prov., Chongan, Sangang, 1979.VI.27, coll. Jikun Yang); (**E**) aedeagus, connective and paramere, dorsal view; (**F**) subgenital plate, dorsal view; (**G**) connective, dorsal view; (**H**) abdominal apodemes, dorsal view; (**I**) male pygofer side, lateral view; (**J**) subgenital plate, dorsal view; (**K**) connective, dorsal view; (**L**) paramere, dorsal view; (**M**) aedeagus, lateral view; and (**N**) apex of aedeagal shaft, dorsal view. Scale bars: 0.5 mm.

**Figure 23 insects-13-00345-f023:**
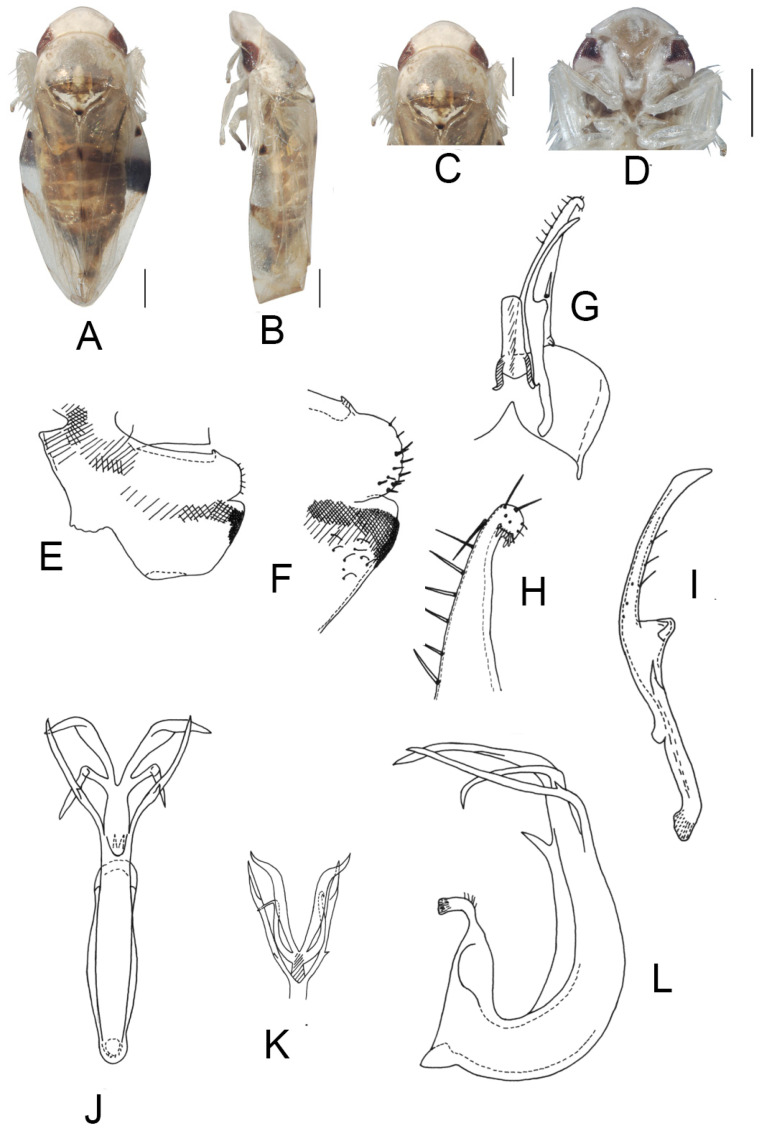
*Eurhadina* (*Singhardina*) *zadyma*
**rec. nov.** (**E**–**L**: Dworakowska, 2002 [3]): (**A**) habitus, dorsal view; (**B**) habitus, lateral view; (**C**) head and thorax, dorsal view; (**D**) face; (**E**) male pygofer side, lateral view; (**F**) hind part of pygofer, lateral view; (**G**) subgenital plate, connective, paramere and sternite IX, dorsal view; (**H**) apex of subgenital plate, dorsal view; (**I**) paramere, dorsal view; (**J**) aedeagus, posterior view; (**K**) apex of aedeagal shaft, dorsal view; and (**L**) aedeagus, lateral view. Scale bars: 0.5 mm.

**Figure 24 insects-13-00345-f024:**
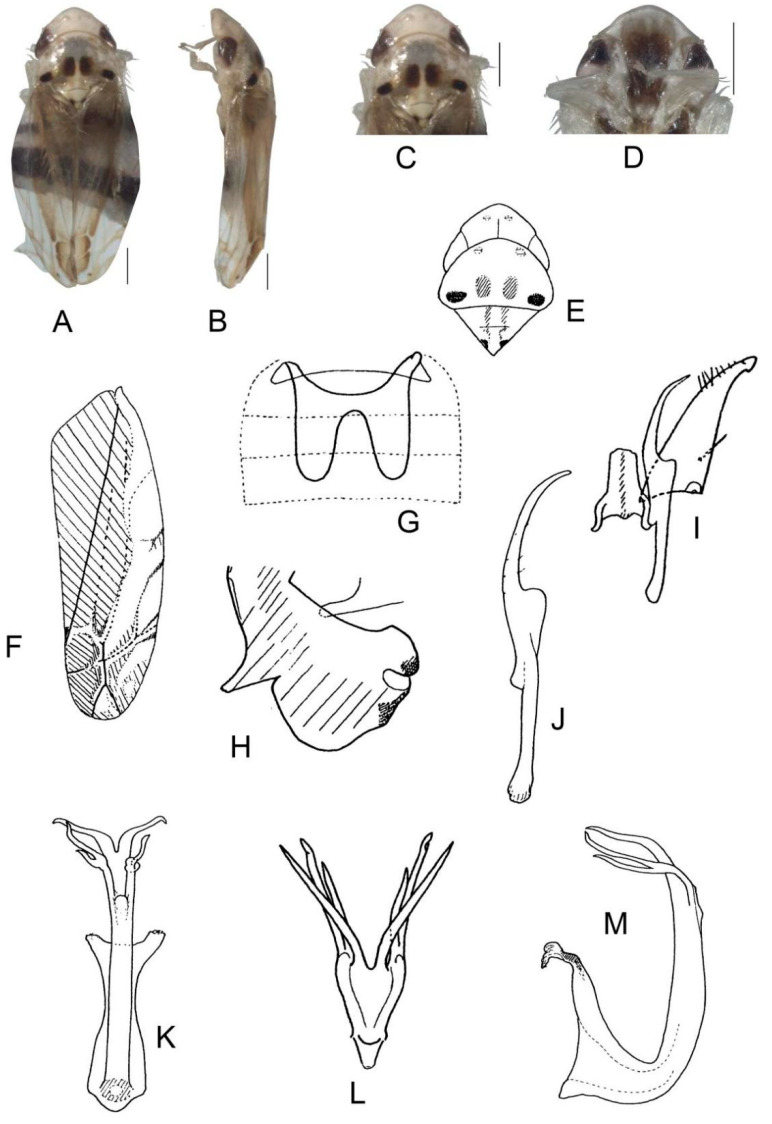
*Eurhadina* (*Singhardina*) *jarrayi*
**rec. nov.** (**E**–**M**: Dworakowska, 2002 [3]): (**A**) habitus, dorsal view; (**B**) habitus, lateral view; (**C**) head and thorax, dorsal view; (**D**) face; (**E**) head and thorax, dorsal view; (**F**) forewing; (**G**) abdominal apodemes, dorsal view; (**H**) pygofer side, lateral view; (**I**) subgenital plate, connective and paramere, dorsal view; (**J**) paramere, dorsal view; (**K**) aedeagus, posterior view; (**L**) apex of aedeagal shaft, dorsal view; and (**M**) aedeagus, lateral view. Scale bars: 0.5 mm.

**Figure 25 insects-13-00345-f025:**
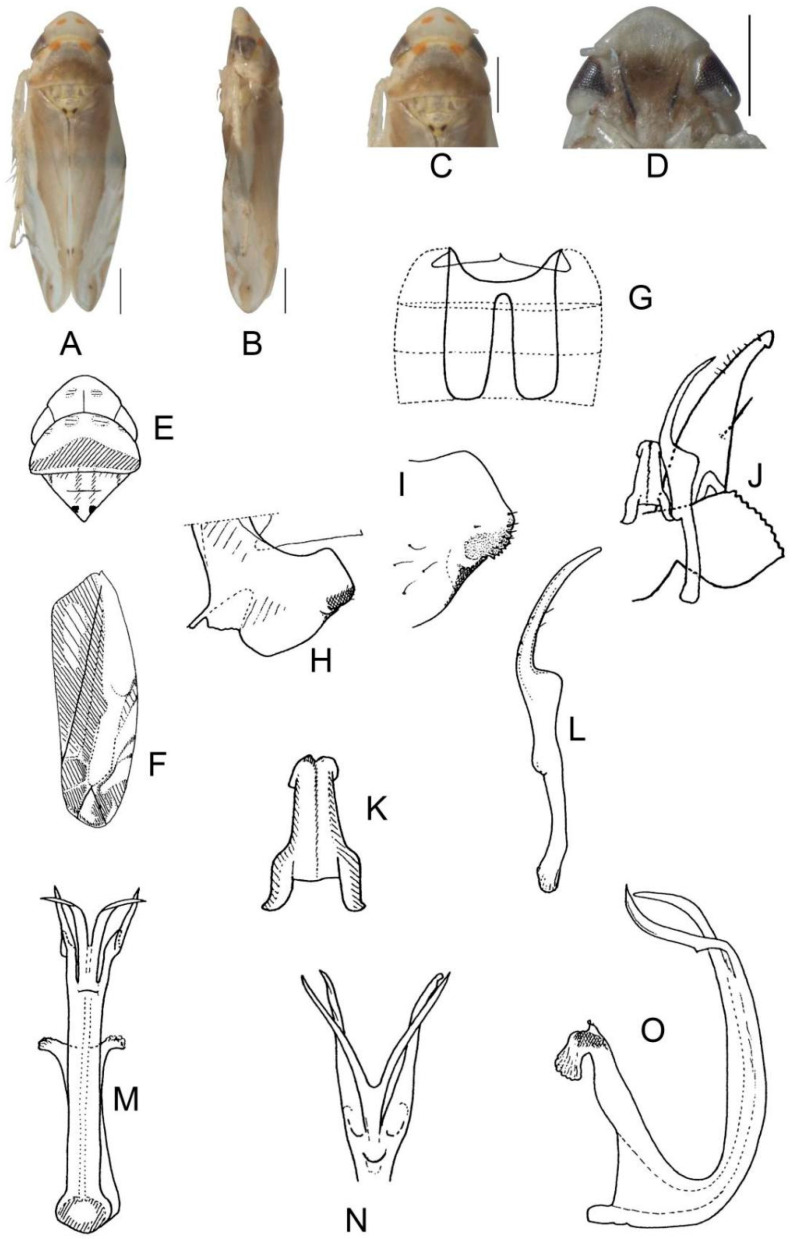
*Eurhadina* (*Singhardina*) *prima*
**rec. nov.** (**E**–**O**: Dworakowska, 2002 [3]): (**A**) habitus, dorsal view; (**B**) habitus, lateral view; (**C**) head and thorax, dorsal view; (**D**) face; (**E**) head and thorax, dorsal view; (**F**) forewing; (**G**) abdominal apodemes, dorsal view; (**H**) pygofer side, lateral view; (**I**) hind part of pygofer, lateral view; (**J**) subgenital plate, connective, paramere and sternite IX, dorsal view; (**K**) connective, dorsal view; (**L**) paramere, dorsal view; (**M**) aedeagus, posterior view; (**N**) apex of aedeagal shaft, dorsal view; and (**O**) aedeagus, lateral view. Scale bars: 0.5 mm.

**Figure 26 insects-13-00345-f026:**
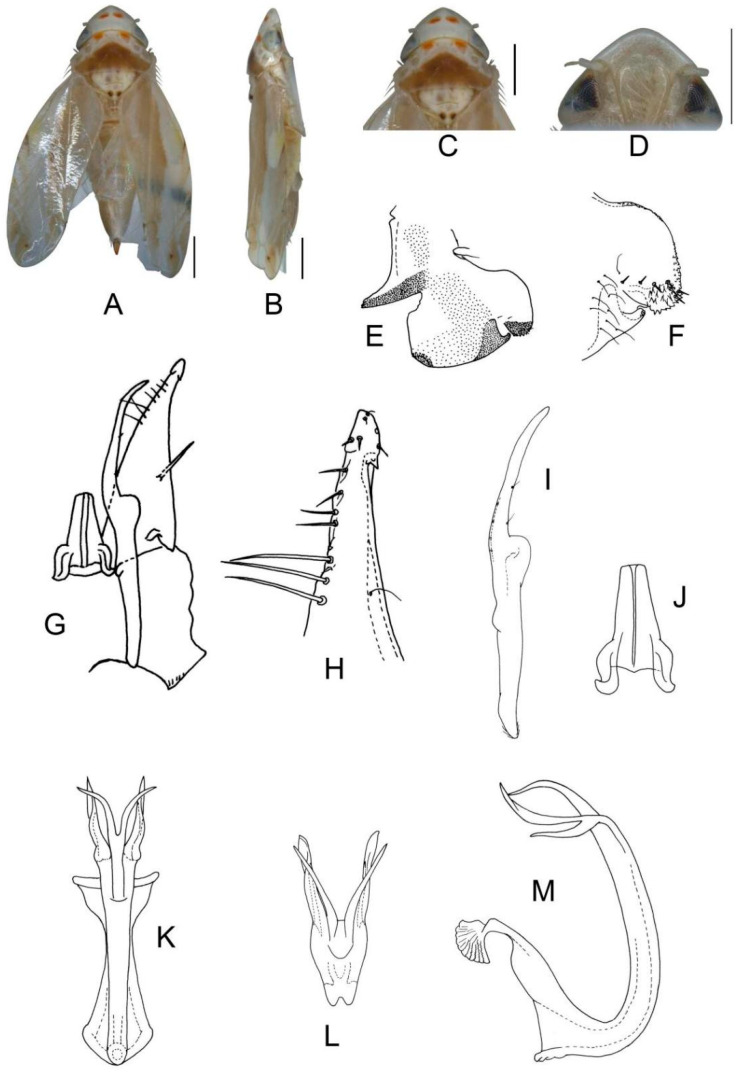
*Eurhadina* (*Singhardina*) *quadrimacularis*
**sp****. nov.**: (**A**) habitus, dorsal view; (**B**) habitus, lateral view; (**C**) head and thorax, dorsal view; (**D**) face; (**E**) pygofer side, lateral view; (**F**) hind part of pygofer, lateral view; (**G**) subgenital plate, connective, paramere and sternite IX, dorsal view; (**H**) apex of subgenital plate, dorsal view; (**I**) paramere, dorsal view; (**J**) connective, dorsal view; (**K**) aedeagus, posterior view; (**L**) apex of aedeagal shaft, dorsal view; and (**M**) aedeagus, lateral view. Scale bars: 0.5 mm.

**Figure 27 insects-13-00345-f027:**
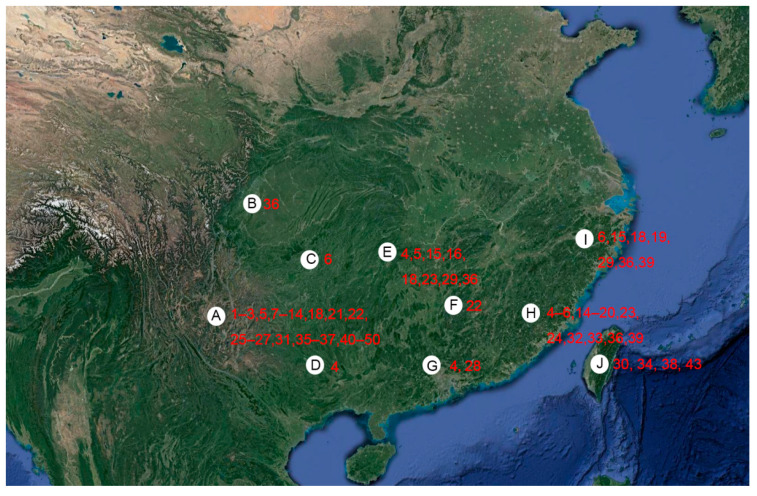
Geographical distribution of *Singhardina* species in China. As shown in the figure: (**A**) Yunnan Prov., (**B**) Sichuan Prov., (**C**) Guizhou Prov., (**D**) Guangxi Prov., (**E**) Hunan Prov., (**F**) Jiangxi Prov., (**G**) Guangdong Prov., (**H**) Fujian Prov., (**I**) Zhejiang Prov., (**J**) Taiwan Prov.; (**1**) *E.* (*S.*) *foliiformis*
**sp. nov.**, (**2**) *E.* (*S.*) *galacta*
**sp. nov.**, (**3**) *E.* (*S.*) *menglunensis*, (**4**) *E.* (*S.*) *recta*
**sp. nov.**, (**5**) *E.* (*S.*) *rubra*, (**6**) *E.* (*S.*) *rubrocorona*, (**7**) *E.* (*S.*) *scalesa*
**sp. nov.**, (**8**) *E.* (*S.*) *scamba*
**sp. nov.**, (**9**) *E.* (*S.*) *scandens*
**sp. nov.**, (**10**) *E.* (*S.*) *unipunctata*, (**11**) *amacularis*
**sp. nov.**, (**12**) *E.* (*S.*) *parilintanonica*
**sp. nov.**, (**13**) *E.* (*S.*) *anurous*, (**14**) *E.* (*S.*) *biavis*, (**15**) *E.* (*S.*) *centralis*, (**16**) *E.* (*S.*) *choui*, (**17**) *E.* (*S.*) *cuii*, (**18**) *E.* (*S.*) *dazhulana*, (**19**) *E.* (*S.*) *diplopunctata*, (**20**) *E.* (*S.*) *exclamationis*, (**21**) *E.* (*S.*) *extensa*
**sp. nov.**, (**22**) *E.* (*S.*) *fasciata*
**rec. nov.**, (**23**) *E.* (*S.*) *flavicorona*, (**24**) *E.* (*S.*) *flaviscutella*
**sp. nov.**, (**25**) *E.* (*S.*) *fusca*, (**26**) *E.* (*S.*) *gracilifurca*
**sp. nov.**, (**27**) *E.* (*S.*) *lata*
**sp. nov.**, (**28**) *E.* (*S.*) *nasti*, (**29**) *E.* (*S.*) *rubrania*, (**30**) *E.* (*S.*) *rubrivittata*, (**31**) *E.* (*S.*) *rutilans*, (**32**) *E.* (*S.*) *spinifera*, (**33**) *E.* (*S.*) *tripunctata*, (**34**) *E.* (*S.*) *unilobata*, (**35**) *E.* (*S.*) *uprotrusa*
**sp. nov.**, (**36**) *E.* (*S.*) *wuyiana*, (**37**) *E.* (*S.*) *zadyma*
**rec. nov.**, (**38**) *E.* (*S.*) *acapitata*, (**39**) *E.* (*S.*) *flavistriata*, (**40**) *E.* (*S.*) *jarrayi*
**rec. nov.**, (**41**) *E.* (*S.*) *prima*
**rec. nov.**, (**42**) *E.* (*S.*) *quadrimacularis*
**sp. nov.**, (**43**) *E.* (*S.*) *yingfengica*, (**44**) *E.* (*S.*) *dissimilis*, (**45**) *E.* (*S.*) *flatilis*, (**46**) *E.* (*S.*) *fumosa*, (**47**) *E.* (*S.*) *furca*, (**48**) *E.* (*S.*) *immatura*, (**49**) *E.* (*S.*) *krispinilla*, and (**50**) *E.* (*S.*) *pookiewica*.

## Data Availability

The data presented in this study are available in this article.

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
