# Peer review of "Eurhadina (Singhardina) Mahmood (Hemiptera: Cicadellidae: Typhlocybinae) from China: A Review of the Asian Species with Descriptions of 14 New Species†"

_insects, 2022, doi:10.3390/insects13040345_

Round 1

Reviewer 1 Report

Most of the comments you can find directly on attached file. Some misspellings must be checked and corrected. PLease check the quality and resolution of drawings copied from other resources. One part is missing in the paper and discussion - the map(s) and even preliminary analysis of biogeographic distribution to support the statement on biodiversity of the group.

Author Response

Point 1: Most of the comments you can find directly on attached file. Some misspellings must be checked and corrected. Please check the quality and resolution of drawings copied from other resources. One part is missing in the paper and discussion - the map(s) and even preliminary analysis of biogeographic distribution to support the statement on biodiversity of the group.

Response 1: Thank you very much for the review. According to your suggestion, we’ve cleaned and upgraded the drawing. Moreover, we have added a map about geographical distribution of Singhardina in China and briefly analyzed its distribution characteristics. We believe that these suggestions can greatly improve this manuscript.

Point 2: Replace "spp." with "sp." (Line 14)

Response 2: Revised as suggested.

Point 3: Which is the reason of use very traditional and disputable term Auchenorrhyncha? The monophyly of this group was discussed several times and still no conclusive results. (Line 24)

Response 3: Many thanks for your comments. Considering the reviewers' advice and to avoid controversy, we delete "(Hemiptera: Auchenorrhyncha)". 

Point 4: Replace "Species" with "species" (Line 60)

Response 4: Sorry for this clerical error, that mistakes have been corrected in the manuscript.

Point 5: Delete "rev.nov"(here and following, it is not part of taxonomy, Line 119)

Response 5: Thanks for your suggestion. We have corrected them in the Key. 

Lines 120, 153, 177, 186: Delete "rev.nov"

Point 6: Are you sure it is a good distinguishing feature or an artifact of preservation/coloration? (Line 220)

Response 6: Yes, we have checked all specimens of E. (S.) foliiformis and E. (S.) rubra kept in our collection and were sure that the former with broader oblique stripe along distal margin of brochosome area than the latter.

Point 7: italics (Line 475)

Response 7: Thanks for pointing this out, we have corrected this mistakes.

Point 8: clean and upgrade this image (Line 544)

Response 8: Thanks for this suggestion. The image was cleaned and upgraded.

Reviewer 2 Report

The manuscript is original and relevant because it carries out a revision of this group of leafhoppers which had to be revised, especially in the light of the new descriptions that have arisen. Furthermore, the last general paper on this same subject dates back to over 50 years ago.

It adds a considerable number of new species and a key to their classification.

The references are appropriate.
Images are fine and clear, drawings are very useful to check species diversity.

The manuscript is well done and can be published as it is. 

Author Response

Point 1: The manuscript is original and relevant because it carries out a revision of this group of leafhoppers which had to be revised, especially in the light of the new descriptions that have arisen. Furthermore, the last general paper on this same subject dates back to over 50 years ago.

It adds a considerable number of new species and a key to their classification.

The references are appropriate.

Images are fine and clear, drawings are very useful to check species diversity.

The manuscript is well done and can be published as it is.

Response 1: Thank you very much for taking the time to read my manuscript and affirming my work.

Reviewer 3 Report

This paper reviewed 50 species of Eurhadina (Singhardina) Mahmood 10 from China, including 14 species, four new records in China, and two synonymies. It is a good study about leafhoppers from China with lots of nice plates. I have made some minor grammatical/typo corrections in your manuscript and some suggestions about adding bars and the length represented by the bars. All of them are in the comments of attached manuscript.

Author Response

Point 1: This paper reviewed 50 species of Eurhadina (Singhardina) Mahmood from China, including 14 species, four new records in China, and two synonymies. It is a good study about leafhoppers from China with lots of nice plates. I have made some minor grammatical/typo corrections in your manuscript and some suggestions about adding bars and the length represented by the bars. All of them are in the comments of attached manuscript.

Response 1: Many thanks for your favorable comments. We’ve addressed your concerns point by point. 

Point 2: It would be better to replace using with under, because you draw it by hand but microscope. This sentence will be changed to Line diagrams of the male genitalia were drawn under an OLYMPUS PM-10AD microscope (Line 53).

Response 2: Thanks for this suggestion. We have replace "using" with "under".

Point 3: Please remove the verb be, because all the descriptions are in telegraphic style. Check all discussion part.

Response 3: Thanks for pointing this out, we have corrected all mistakes throughout the paper.

Line 69: Delete "be"

Line 944: Delete "tend to be"

Point 4: How many millimeters do this bars represent? Please note them.

Response 4: Thanks for your suggestion. We’ve supplied this information in all the captions.

Lines 246, 278, 289, 320, 334, 350, 379, 409, 439, 487, 518, 566, 587, 631, 669, 687, 699, 728, 763, 796, 823, 838, 879, 905, 926, 967, 981, 1015: added "Scale bars: 0.5 mm."

Point 5: It would be better to add some bars for these drawing.

Response 5: This really is a valuable suggestion. But, those figures were drawn under a microscope without the ruler. So, we are sorry, it’s hard for us to measure it and add bars for these drawing.